# Fairness with Overlapping Groups

**Forest Yang**[*]
UC Berkeley

**Moustapha Cisse**
Google Research Accra

**Sanmi Koyejo**
Google Research Accra & Illinois

## Abstract

In algorithmically fair prediction problems, a standard goal is to ensure the equality of fairness metrics across multiple overlapping groups simultaneously. We reconsider this standard fair classification problem using a probabilistic population analysis, which, in turn, reveals the Bayes-optimal classifier. Our approach unifies a variety of existing group-fair classification methods and enables extensions to a wide range of non-decomposable multiclass performance metrics and fairness measures. The Bayes-optimal classifier further inspires consistent procedures for algorithmically fair classification with overlapping groups. On a variety of real datasets, the proposed approach outperforms baselines in terms of its fairness-performance tradeoff.

## 1 Introduction

Machine learning informs an increasingly large number of critical decisions in diverse settings. They assist medical diagnosis [17], guide policing [18], and power credit scoring systems [23]. While they have demonstrated their value in many sectors, they are prone to unwanted biases, leading to discrimination against protected subgroups within the population. For example, recent studies have revealed biases in predictive policing and criminal sentencing systems [18, 6]. The blossoming body of research in algorithmic fairness aims to study and address this issue by introducing novel algorithms guaranteeing a certain level of non-discrimination in the predictions. Each such algorithm relies on a specific definition of fairness, which falls into one of two categories: Individual fairness [9, 26] or group fairness [4, 14, 11]. The vast majority of the algorithmic group fairness literature has focused on the simplest case where there are only two groups. In this paper, we consider the more nuanced case of group fairness with respect to multiple groups.

The simplest setting is the *independent* case, with only one sensitive attribute which can take multiple values, e.g., race only. The presence of multiple sensitive attributes (e.g., race *and* gender simultaneously) leads to non-equivalent definitions of group fairness. On the one hand, fairness can be considered independently per sensitive attribute, leading to overlapping subgroups. For example, consider a model restricted to demographic parity between subgroups defined by ethnicity. Simultaneously, the model can be constrained to fulfill demographic parity between subgroups defined by gender. We term fairness in this situation *independent group fairness*. On the other hand, one can consider all subgroups defined by intersections of sensitive attributes (e.g., ethnicity and gender), leading to *intersectional group fairness*. A given algorithm can be *independently group fair*, e.g., when considering race and gender in isolation, but not *intersectionally group fair*, e.g., when considering intersections of racial and gender groups. For example, Buolamwini and Gebru [3], showed how facial recognition software had a particularly poor performance for black women. This phenomenon, called *fairness gerrymandering*, has been studied by Kearns et al. [15]. Intersectional

---

[*]Work completed while an intern at Google Accra.

fairness is often considered ideal. However, it comes with major statistical and computational hurdles such as data scarcity at intersections of minority groups, and the potentially exponential number of subgroups. Indeed, current algorithms consist of either brute force enumeration or searching via a cost-sensitive classification problem, and intersectional groups are often empty with finite samples [15]. On the other hand, independent group fairness still provides a broad measure of fairness and is much easier to enforce.

We seek to *design unifying statistically consistent strategies for group fairness and to clarify the relationship between the existing definitions.* Our main results and algorithms apply to arbitrary overlapping group definitions. Our contributions are summarized in the following.

- **Probabiistic results**. We characterize the population optimal (also known as the Bayes-optimal) prediction procedure for multiclass classification, where all the metrics are general linear functions of the confusion matrix. We consider both overlapping (independent, gerrymandering) and non-overlapping (unrestricted, intersectional) group fairness.
- **Algorithms and statistical results.** Inspired by the population optimal, we propose simple plugin and weighted empirical risk minimization (ERM) approaches for algorithmically fair classification, and prove their consistency, i.e., the empirical estimator converges to the population optimal with sufficiently large samples. Our general approach recovers existing results for plugin and weighted ERM group-fair classifiers.
- **Comparisons.** We compare independent group fairness to the overlapping case. We show that intersectional fairness implies overlapping group fairness under weak conditions. However, the converse is not true, i.e., overlapping fairness may not imply intersectional fairness. This result formalizes existing observations on the dangers of gerrymandering.
- **Evaluation.** Empirical results are provided to highlight our theoretical claims.

Taken together, our results unify and advance the state of the art with respect to the probabilistic, statistical, and algorithmic understanding of group-fair classification. The generality of our approach gives significant flexibility to the algorithm designer when constructing algorithmically-fair learners.

## 2  Problem Setup and Notation

Throughout the paper, we use uppercased bold letters to represent matrices, and lowercased bold letters to represent vectors. Let $e_i$ represent the $i$th standard basis whose $i$th dimension is 1 and 0 otherwise $e_i = (0, \cdots, 1, \cdots, 0)$. We denote $\mathbf{1}$ as the all-ones vector with dimension inferred from context. Given two matrices $\mathbf{A}, \mathbf{B}$ of same dimension, $\langle \mathbf{A}, \mathbf{B} \rangle = \sum_{i,j} a_{ij} b_{ij}$ is the Frobenius inner product. For any quantity $q$, $\hat{q}$ denotes an empirical estimate. Due to limited space, proofs are presented in the appendix.

**Group notation.** We assume $M$ sensitive attributes, where each attribute is indicated by a group $\{\mathcal{A}_m\}_{m \in [M]}$. For example, $\mathcal{A}_1$ may correspond to race, $\mathcal{A}_2$ may correspond to gender, and so on. Combined, the sensitive group indicator is represented by a $M$-dimensional vector $\mathbf{a} \in \mathcal{A} = \mathcal{A}_1 \times \mathcal{A}_2 \times \cdots \mathcal{A}_M$. In other words, each instance is associated with $M$ subgroups simultaneously.

**Probabilistic notation.** Consider the multiclass classification problem where $\mathcal{Z}$ denotes the instance space and $\mathcal{Y} = [K]$ denotes the output space with $K$ classes. We assume the instances, outputs and groups are samples from a probability distribution $\mathbb{P}$ over the domain $\mathcal{Y} \times \mathcal{Z} \times \mathcal{A}$. A dataset is given by $n$ samples $(y^{(i)}, z^{(i)}, a^{(i)}) \overset{\text{i.i.d}}{\sim} \mathbb{P}, i \in [n]$. To simplify notation, let $\mathcal{X} = \mathcal{Z} \times \mathcal{A}$, so $\mathbf{x} = (\mathbf{z}, \mathbf{a})$. Define the set of randomized classifiers $\mathcal{H}_r = \{\mathbf{h} : \mathcal{X} \times \mathcal{A} \rightarrow (\Delta^K)\}$, where $\Delta^q = \{\mathbf{p} \in [0,1]^q : \sum_{i=1}^q p_i = 1\}$ is the $q-1$ dimensional probability simplex. A classifier $\boldsymbol{h}$ is associated with the random variable $h \in [K]$ defined by $\mathbb{P}(h = k | \mathbf{x}) = h_k(\mathbf{x})$. If $\boldsymbol{h}$ is deterministic, then we can write $\boldsymbol{h}(\mathbf{x}) = e_{h(\mathbf{x})}$.

*Confusion matrices.* For any multiclass classifier, let $\boldsymbol{\eta}(\mathbf{x}) \in \Delta^K$ denote the class probabilities for any given instance $\mathbf{x}$ and sensitive attribute $\mathbf{a}$, whose $k$th element is the conditional probability of the output belonging to class $k$, i.e., $\eta_k(\mathbf{x}) = \mathbb{P}(Y_m = k \mid X = \mathbf{x})$. The population confusion matrix is $\mathbf{C} \in [0,1]^{K \times K}$, with elements defined for $k, \ell \in [K]$ as $\mathbf{C}_{k,\ell} = \mathbb{P}(Y = k, h = \ell)$, or equivalently,

$$\mathbf{C}_{k,\ell} = \int_{\mathbf{x}} \boldsymbol{\eta}_k(\mathbf{x}) h_\ell(\mathbf{x}) \, d\mathbb{P}(\mathbf{x}).$$

*Group-specific confusion matrices.* Let $\mathcal{G}$ represent a set of subsets of the instances, i.e., potentially overlapping partitions of the instances $\mathcal{X}$. We leave $\mathcal{G}$ as generic for now, and will specify cases specific to fairness in the following. Given any group $g \in \mathcal{G}$, we can define the group-specific confusion matrix $\mathbf{C}^g \in [0,1]^{K \times K}$, with elements defined for $k, \ell \in [K]$, where

$$\mathbf{C}^g_{k,\ell} = \int_{\mathbf{x}} \boldsymbol{\eta}_k(\mathbf{x}) h_\ell(\mathbf{x}) \, d\mathbb{P}(\mathbf{x}|\mathbf{x} \in g).$$

We will abbreviate the event $\{\mathbf{x} \in g\}$ to simply $g$ when it is clear from context. Let $\pi_g = \mathbb{P}(X \in g)$ be the probability of group $g$. It is clear that when the groups $\mathcal{G}$ form a partition, i.e., $a \cap b = \emptyset \; \forall a, b \in \mathcal{G}$ and $\bigcup_{g \in \mathcal{G}} g = \mathcal{X}$, the population confusion may be recovered by a weighted average of group confusions, $\mathbf{C} = \sum_{g \in \mathcal{G}} \pi_g \mathbf{C}^g$.

**The sample confusion matrix** is defined as $\widehat{\mathbf{C}}[\boldsymbol{h}] = \frac{1}{n} \sum_{i=1}^n \widehat{\mathbf{C}}^{(i)}[\boldsymbol{h}]$, where $\widehat{\mathbf{C}}^{(i)}[\boldsymbol{h}] \in [0,1]^{K \times K}$, and $\widehat{C}^{(i)}_{k,\ell}[\boldsymbol{h}] = [\![y_i = k]\!] h_\ell(\mathbf{x}_i)$. Here, $[\![\cdot]\!]$ is the indicator function, so $\sum_{k=1}^K \sum_{\ell=1}^K \widehat{C}^{(i)}_{k,\ell}[\boldsymbol{h}] = 1$. *The empirical group-specific confusion matrices* $\widehat{\mathbf{C}}^g$ are computed by conditioning on groups. In the empirical case, it is convenient to represent group memberships via indices alone, i.e., $\mathbf{x}_i \in g$ as $i \in g$. We have $\widehat{\mathbf{C}}^g[\boldsymbol{h}] = \frac{1}{|g|} \sum_{i \in g} \widehat{\mathbf{C}}^{(i)}[\boldsymbol{h}]$.

**Fairness constraints.** Let $\mathcal{G}_{\text{fair}}$ represent the (potentially overlapping) set of groups across which we wish to enforce fairness. The following states our formal assumptions on $\mathcal{G}_{\text{fair}}$.

**Assumption 2.1.** $\mathcal{G}_{\text{fair}}$ is a function of the sensitive attributes $\mathcal{A}$ only.

We will focus the discussion on common cases in the literature. These include non-overlapping (unrestricted, intersectional), and overlapping (independent, gerrymandering) group partitions. They are computed for an example in figure 1.

- *Unrestricted case.* The simplest case is where the group is defined by a single sensitive attribute (when there are multiple sensitive attributes, all but one are ignored). These have been the primary settings addressed by past literature [11, 20, 1]. Thus for some fixed $i \in [M]$, $g_j = \{(\mathbf{z}, \mathbf{a}) | a_i = j\}$, so $|\mathcal{G}_{\text{unrestricted}}| = |A_i|$. In the special case of binary sensitive attributes, $|\mathcal{G}_{\text{unrestricted}}| = 2$.
- *Intersectional groups.* Here, the non-overlapping groups are associated with all possible combinations of sensitive features. Thus $g_{\mathbf{a}} = \{(\mathbf{z}, \mathbf{a}') | \mathbf{a}' = \mathbf{a}\} \; \forall \mathbf{a} \in \mathcal{A}$ so $|\mathcal{G}_{\text{intersectional}}| = \prod_{m \in M} |\mathcal{A}_m|$. In the special case of binary sensitive attributes, $|\mathcal{G}_{\text{intersectional}}| = 2^M$.
- *Independent groups.* Here, the groups are overlapping, with a set of groups associated with each fairness attribute separately. It is convenient to denote the groups based on indices representing each attribute, and each potential setting. Thus $g_{i,j} = \{(\mathbf{z}, \mathbf{a}) | a_i = j\}$, so $|\mathcal{G}_{\text{independent}}| = \sum_{m \in M} |\mathcal{A}_m|$. In the special case of binary sensitive attributes, $|\mathcal{G}_{\text{independent}}| = 2M$.
- *Gerrymandering intersectional groups.* Here, group intersections are defined by any subset of the sensitive attributes, leading to overlapping subgroups. $\mathcal{G}_{\text{gerrymandering}} = \{\{(\mathbf{z}, \mathbf{a}) : \mathbf{a}_I = \mathbf{s}\} : I \subset [M], \mathbf{s} \in \mathcal{A}_I\}$ where $\mathbf{a}_I$ denotes $\mathbf{a}$ restricted to the entries indexed by $I$. It is also the closure of $\mathcal{G}_{\text{independent}}$ under intersection. As a result, $\mathcal{G}_{\text{intersectional}} \subseteq \mathcal{G}_{\text{gerrymandering}}$, and $\mathcal{G}_{\text{independent}} \subseteq \mathcal{G}_{\text{gerrymandering}}$. In the special case of binary sensitive attributes, $|\mathcal{G}_{\text{gerrymandering}}| = 3^M$.

**Fairness metrics.** We formulate group fairness by upper bounding a fairness violation function $\mathcal{V} : \mathcal{H} \mapsto \mathbb{R}^J$ which can be represented as a linear function of the confusion matrices, i.e. $\mathcal{V}(\boldsymbol{h}) = \Phi(\mathbf{C}[\boldsymbol{h}], \{\mathbf{C}^g[\boldsymbol{h}]\}_{g \in \mathcal{G}_{\text{fair}}})$ where $\forall j \in [J]$, $\mathcal{V}(\boldsymbol{h})_j = \phi_j(\mathbf{C}[\boldsymbol{h}], \{\mathbf{C}^g[\boldsymbol{h}]\}_{g \in \mathcal{G}_{\text{fair}}}) = \langle \mathbf{U}_j, \mathbf{C} \rangle - \sum_{g \in \mathcal{G}_{\text{fair}}} \langle \mathbf{V}_j^g, \mathbf{C}^g \rangle$. This formulation is sufficiently flexible to include the fairness statistics we are aware of in common use as special cases. For example, demographic parity for binary classifiers [9] can be defined by fixing $\mathbf{C}^g_{0,0} + \mathbf{C}^g_{1,1}$ across groups. Equal opportunity [12] is recovered by fixing the group-specific true positives, using population specific weights, i.e.,

$$\phi_{\text{DP}}^{\pm} = \pm(\mathbf{C}^g_{0,1} + \mathbf{C}^g_{1,1} - \mathbf{C}_{0,1} + \mathbf{C}_{1,1}) - \nu, \quad \phi_{\text{EO}}^{\pm} = \pm \left( \frac{1}{\mathbb{P}(Y = 1 \mid g)} \mathbf{C}^g_{1,1} - \frac{1}{\mathbb{P}(Y = 1)} \mathbf{C}_{1,1} \right) - \nu,$$

using both a positive and negative constraint to penalize both positive and negative deviations between the group and the population, and relaxation $\nu$.

**Performance metrics.** We consider an error metric $\mathcal{E} : \mathcal{H} \mapsto \mathbb{R}_+$ that is a linear function of the population confusion $\mathcal{E}(\mathbf{h}) = \psi(\mathbf{C}) = \langle \mathbf{D}, \mathbf{C}[\boldsymbol{h}] \rangle$. This setting has been studied in binary

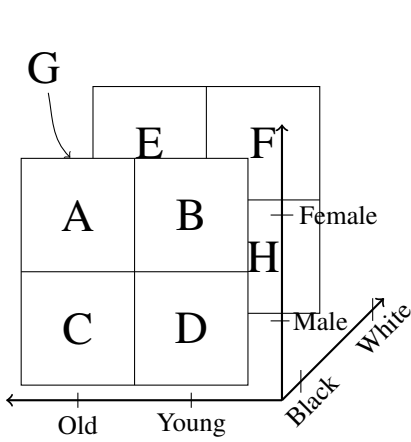

Independent groups:
$$\{\text{Female} = A \cup B \cup E \cup F, \text{ Male} = C \cup D \cup G \cup H,$$
$$\text{Old} = A \cup C \cup E \cup G, \text{ Young} = B \cup D \cup F \cup H,$$
$$\text{Black} = A \cup B \cup C \cup D, \text{ White} = E \cup F \cup G \cup H\}$$

Intersectional groups:
$$\{A, B, C, D, E, F, G, H\}$$

Gerrymandering groups:
$$\{\text{Female, Male, Old, Young, Black, White,}$$
$$\text{Fe \&} \{\text{Od, Bk, Yg, Wt}\} = A \cup E, A \cup B, B \cup F, E \cup F,$$
$$\text{Me \&} \{\text{Od, Bk, Yg, Wt}\} = C \cup G, C \cup D, D \cup H, G \cup H,$$
$$\text{Od \&} \{\text{Bk, Wt}\} = A \cup C, E \cup G,$$
$$\text{Yg \&} \{\text{Bk, Wt}\} = B \cup D, F \cup H,$$
$$A, B, C, D, E, F, G\}$$

Figure 1: Independent, intersectional and gerrymandering intersectional group fairness refer to satisfying equality of a certain metric across prescribed sets of groups at various levels of granularity, shown above for an example with $M = 3$ protected attributes corresponding to sex, age, and race.

classification [25], multiclass classification [21], multilabel classification [16], and multioutput classification [24]. For instance, standard classification error corresponds to setting $\mathbf{D} = 1 - \mathbf{I}$. The goal is to learn the Bayes-optimal classifier with respect to the given metric, which, when it exists, is given by:

$$\boldsymbol{h}^* \in \operatorname{argmin}_{\boldsymbol{h}} \ \mathcal{E}(\boldsymbol{h}) \text{ s.t. } \mathcal{V}(\boldsymbol{h}) \leq \mathbf{0}. \tag{1}$$

We denote the optimal error as $\mathcal{E}^* = \mathcal{E}(\boldsymbol{h}^*)$. We say a classifier $\boldsymbol{h}_N$ constructed using finite data of size $N$ is $\{\mathcal{E}, \mathcal{V}\}$-consistent if $\mathcal{E}(\boldsymbol{h}_n) \xrightarrow{\mathbb{P}} \mathcal{E}^*$ and $\mathcal{V}(\boldsymbol{h}_n) \xrightarrow{\mathbb{P}} \mathbf{0}$, as $n \to \infty$. We also consider empirical versions of error $\widehat{\mathcal{E}}(\boldsymbol{h}) = \psi(\widehat{\mathbf{C}}[\boldsymbol{h}])$ and fairness violation $\widehat{\mathcal{V}}(\boldsymbol{h}) = \Phi(\widehat{\mathbf{C}}[\boldsymbol{h} \{\widehat{\mathbf{C}}^g[\boldsymbol{h}]\}_g)$.

## 3 Bayes-Optimal Classifiers

In this section, we identify a parametric form for the Bayes-optimal group-fair classifier under standard assumptions. To begin, we introduce the following general assumption on the joint distribution.

**Assumption 3.1** ($\eta$-continuity). Assume $\mathbb{P}(\{\boldsymbol{\eta}(\mathbf{x}) = \mathbf{c}\}) = 0 \ \forall \mathbf{c} \in \Delta^K$. Furthermore, let $Q = \boldsymbol{\eta}(\mathbf{x})$ be a random variable with density $p_\eta(Q)$, where $p_\eta(Q)$ is absolutely continuous with respect to the Lebesgue measure restricted to $\Delta^K$.

This assumption imposes that the conditional probability as a random variable has a well-defined density. Analogous regularity assumptions are widely employed in literature on designing well-defined complex classification metrics and seem to be unavoidable (we refer interested reader to Yan et al. [25], Narasimhan et al. [21] for details). Next, we define the general form of weighted multiclass classifiers, which are the Bayes-optimal classifiers for linear metrics.

**Definition 3.2.** [Narasimhan et al. [21]] Given a loss matrix $\mathbf{W} \in \mathbf{R}^{K \times K}$, a weighted classifier $\mathbf{h}$ satisfies $h_i(\mathbf{x}) > 0$ only if $i \in \arg\min_{k \in [K]} \langle \mathbf{W}_k, \eta(\mathbf{x}) \rangle$.

Next we present our first main result identifying the Bayes-optimal group-fair classifier.

**Theorem 3.1.** *Under Assumption 2.1 and Assumption 3.1, if (1) is feasible (i.e., a solution exists), the Bayes-optimal classifier is given by $\mathbf{h}^*(\mathbf{x}) = \mathbf{h}^*(\mathbf{z}, \mathbf{a}) = \beta_{\mathbf{a}}\mathbf{h}_1(\mathbf{x}) + (1 - \beta_{\mathbf{a}})\mathbf{h}_2(\mathbf{x})$, where $\beta_{\mathbf{a}} \in (0, 1), \forall \mathbf{a} \in \mathcal{A}$ and $\mathbf{h}_i(\mathbf{x})$ are weighted classifiers with weights $\{\{\mathbf{W}_{i,\mathbf{a}}\}_{i \in \{1,2\}}\}_{\mathbf{a} \in \mathcal{A}}$.*

One key observation is that pointwise, the Bayes-optimal classifier can be decomposed based on intersectional groups $\mathcal{G}_{\text{intersectional}} = \mathcal{A}$, even when $\mathcal{G}_{\text{fair}}$ is overlapping. This observation will prove useful for algorithms.

### 3.1 Intersectional group fairness implies overlapping group fairness

Recent research [15] has shown how imposing overlapping group fairness using independent fairness restrictions can lead to violation of intersectional fairness, primarily via examples. This observation

**Algorithm 1:** `GroupFair`, Group-fair classification with overlapping groups,

**Input:** $\psi : [0,1]^{K \times K} \to [0,1]$, $\Phi : [0,1]^{K \times K} \times ([0,1]^{K \times K})^{\mathcal{G}_{\text{fair}}} \to [0,1]^J$
samples $\{(\mathbf{x}_1, y_1), \dots, (\mathbf{x}_n, y_n)\}$.
Initialize $\boldsymbol{\lambda}_1 \in [0, B]^J$;
**for** $t = 1, \dots, T$ **do**
$\quad \left| \begin{array}{l} h^t \leftarrow \text{MinOracle}_{h \in \mathcal{H}}(\mathcal{L}(h, \boldsymbol{\lambda}^t), z^n); \\ \boldsymbol{\lambda}^{t+1} \leftarrow \text{Update}_t(\boldsymbol{\lambda}^t, \Phi(\widehat{\mathbf{C}}[h^t], \{\widehat{\mathbf{C}}^g[h^t]\}_{g \in \mathcal{G}_{\text{fair}}}) - \varepsilon); \end{array} \right.$
**end**
$\bar{\boldsymbol{h}}^T \leftarrow \frac{1}{T} \sum_{t=1}^{T} \boldsymbol{h}^t, \quad \bar{\boldsymbol{\lambda}}^T \leftarrow \frac{1}{T} \sum_{t=1}^{T} \boldsymbol{\lambda}^t;$
**return** $(\bar{\boldsymbol{h}}^T, \bar{\boldsymbol{\lambda}}^T)$

led to the term *fairness gerrymandering*. Here, we examine this claim more formally, showing that enforcing intersectional fairness controls overlapping fairness, although the converse is not always true, i.e., enforcing overlapping fairness does not imply intersectional fairness. We show this result for the general case of quasi-convex fairness measures, with linear fairness metrics recovered as a special case.

**Proposition 3.2.** *For any $\mathcal{G}_{fair}$ that satisfies assumption 2.1, suppose $\phi : [0,1]^{K \times K} \times [0,1]^{K \times K} \to \mathbb{R}_+$ is quasiconvex, $\phi(\mathbf{C}, \mathbf{C}^g) \leq 0 \, \forall g \in \mathcal{G}_{intersectional} \implies \phi(\mathbf{C}, \mathbf{C}^g) \leq 0 \, \forall g \in \mathcal{G}_{fair}$. The converse does not hold.*

**Remark 3.3.** *Note that the converse claim of Proposition 3.2, does not apply to $\mathcal{G}_{gerrymandering}$. Controlling the gerrymandering fairness violation implies control of the intersectional fairness violation, since $\mathcal{G}_{intersectional} \subseteq \mathcal{G}_{gerrymandering}$.*

## 4 Algorithms

Here we present `GroupFair`, a general empirical procedure for solving (1). The Lagrangian of the constrained optimization problem (1) is $\mathcal{L}(\boldsymbol{h}, \boldsymbol{\lambda}) = \mathcal{E}(\boldsymbol{h}) + \boldsymbol{\lambda}^\top \mathcal{V}(\boldsymbol{h})$ with empirical Lagrangian $\hat{\mathcal{L}}(\boldsymbol{h}, \boldsymbol{\lambda}) = \hat{\mathcal{E}}(\boldsymbol{h}) + \boldsymbol{\lambda}^\top (\mathcal{V}(\boldsymbol{h}) - \varepsilon)$, where $\varepsilon$ is a buffer which allows us to compare the loss of the computed predictor to that of the optimal fair predictor, when proving generalization properties of our algorithm.

Our approach involves finding a saddle point of the Lagrangian by alternating between computing the minimizer of the current Lagrangian, and updating the Lagrange multipliers based on the current fairness violations. The returned classifiers will be probabilistic combinations of classifiers in $\mathcal{H}$, i.e. the procedure returns a classifier in $\text{conv}(\mathcal{H})$. In the following, we first assume the dual parameter $\boldsymbol{\lambda}$ is fixed, and describe ways of computing the minimization by calling standard classification algorithms.

We consider both plugin and weighted ERM. In brief, the plugin estimator first proceeds assuming $\boldsymbol{\eta}(\mathbf{x})$ is known, then we *plugin* the empirical estimator $\hat{\boldsymbol{\eta}}(\mathbf{x})$ in its place. The plugin approach has the benefit of low computational complexity once fixed. On the other hand, the weighted ERM estimator requires the solution of a weighted classification problem in each round, but avoids the need for estimating $\hat{\boldsymbol{\eta}}(\mathbf{x})$.

### 4.1 Weighted ERM Oracle

In the weighed ERM approach we parametrize $h : \mathcal{X} \to [K]$ by a function class $\mathcal{F}$ of functions $\mathbf{f} : \mathcal{X} \to \mathbb{R}^K$. The classification is the argmax of the predicted vector, $h(\mathbf{x}) = \text{argmax}_j(\mathbf{f}(\mathbf{x})_j)$, so we denote the set of classifiers as $\mathcal{H}^{werm} = \text{argmax} \circ \mathcal{F}$. The following special case of Definition 1 in [22] outlines the required conditions for weighted multiclass classification calibration. This is commonly referred to as cost-sensitive classification [1] when applied to binary classification.

**Definition 4.1** (**W**-calibration [22]). Let $\mathbf{W} \in \mathbb{R}_+^{K \times K}$. A surrogate function $\mathbf{L} : \mathbb{R}^K \to \mathbb{R}_+^K$ is said to be **W**-calibrated if

$$\forall p \in \Delta^K : \inf_{\mathbf{u}:\text{argmax}(u)\notin\text{argmin}_k(\mathbf{p}^\top \mathbf{W})_k} \mathbf{p}^\top \mathbf{L}(\mathbf{u}) > \inf_{\mathbf{u}} \mathbf{p}^\top \mathbf{L}(\mathbf{u}).$$

Note that the weights are sample (group) specific – which, while uncommon, is not new, e.g., Ávila Pires et al. [27].

**Proposition 4.1.** *The weighted ERM estimator for average fairness violation is given by:* $h(\mathbf{x}) = \operatorname{argmax}_j(\mathbf{f}^*(\mathbf{x})_j)$, $\mathbf{f}^* = \min_{\mathbf{f} \in \mathcal{F}} \hat{L}(f)$; *where* $\hat{L}(\mathbf{f}) = \hat{\mathbb{E}}[\mathbf{y}^T \mathbf{L}(\mathbf{f})]$ *is a multiclass classification surrogate for the weighted multiclass error with group-dependent weights* $\forall \mathbf{a} \in \mathcal{A}$

$$\mathbf{W}(\mathbf{x}) = \left[ \mathbf{D} + \sum_{j=1}^{J} \boldsymbol{\lambda}_j \left( \mathbf{U}_j - \sum_{g \in \mathcal{G}_{\text{fair}}} \frac{\mathbb{1}_{\mathbf{a} \in g}}{\hat{\pi}(g)} \mathbf{V}_j^g \right) \right]. \tag{2}$$

## 4.2 The Plugin Oracle

The plugin hypothesis class are the weighted classifiers, identified by Theorem 3.1 as $\mathcal{H}^{plg} = \{h(\mathbf{x}) = \operatorname{argmin}_{j \in [K]} (\hat{\boldsymbol{\eta}}(\mathbf{x})^\top \mathbf{B}(\mathbf{x}))_j : \mathbf{B}(\mathbf{x}) \in \mathbb{R}^{K \times K}\}$. Here, we focus on the average violation case only. By simply-reordering terms, the population problem can be determined as follows.

**Proposition 4.2.** *The plug-in estimator for average fairness violation is given by* $\hat{h}(\mathbf{x}) = \operatorname{argmin}_{k \in [K]} (\boldsymbol{\eta}(\mathbf{x})^\top \mathbf{W}(\mathbf{x}))_k$, *where* $\mathbf{W}(\mathbf{x})$ *is defined in* (2).

## 4.3 `GroupFair`, a General Group-Fair Classification Algorithm

We can now present `GroupFair`, a general algorithm for group-fair classification with overlapping groups, as outlined in Algorithm 1. As outlined, our approach proceeds in rounds, updating the classifier oracle and the dual variable. Interleaved with the primal update is a dual update $\mathrm{Update}_t(\boldsymbol{\lambda}, \mathbf{v})$ via gradient descent on the dual variable. The resulting classifier is the average over the oracle classifiers.

**Recovery of existing methods.** When the groups are non-overlapping, `GroupFair` with the Plugin oracle and projected gradient ascent update recovers FairCOCO [20]. Similarly, when the groups are non-overlapping, and the labels are binary, `GroupFair` with the weighted ERM oracle and exponentiated gradient update recovers FairReduction [1] (see also Table 1). Importantly, `GroupFair` enables a straightforward extension to overlapping groups.

# 5 Consistency

Here we discuss the consistency of the weighted ERM and the plugin approaches. For any class $\mathcal{H} = \{h : \mathcal{X} \to [K]\}$, denote $\mathcal{H}_k = \{\mathbb{1}_{\{h(x)=k\}} : h \in \mathcal{H}\}$. We assume WLOG that $\mathrm{VC}(\mathcal{H}_1) = \ldots = \mathrm{VC}(\mathcal{H}_K)$ and denote this quantity as $\mathrm{VC}(\mathcal{H})$. Next, we give a theorem relating the performance and satisfaction of constraints of an empirical saddle point to an optimal fair classifier.

**Theorem 5.1.** *Suppose* $\psi : [0,1]^{K \times K} \to [0,1]$ *and* $\Phi : [0,1]^{K \times K} \times ([0,1]^{K \times K})^{\mathcal{G}_{\text{fair}}} \to [0,1]^J$ *are $\rho$-Lipschitz w.r.t.* $\|\cdot\|_\infty$. *Recall* $\hat{\mathcal{L}}(\boldsymbol{h}, \boldsymbol{\lambda}) = \hat{\mathcal{E}}(\boldsymbol{h}) + \boldsymbol{\lambda}^\top (\hat{\mathcal{V}}(\boldsymbol{h}) - \varepsilon \mathbf{1})$. *Define* $\gamma(n', \mathcal{H}, \delta) = \sqrt{\frac{\mathrm{VC}(\mathcal{H}) \log(n) + \log(1/\delta)}{n}}$. *If* $n_{\min} = \min_{g \in \mathcal{G}_{\text{fair}}} n_g$, $\varepsilon = \Omega(\rho \gamma(n_{\min}, \mathcal{H}, \delta))$ *then w.p.* $1 - \delta$:

*If* $(\bar{\boldsymbol{h}}, \bar{\boldsymbol{\lambda}})$ *is a $\nu$-saddle point of* $\max_{\boldsymbol{\lambda} \in [0,B]^J} \min_{\boldsymbol{h} \in \mathrm{conv}\,\mathcal{H}} \hat{\mathcal{L}}(\boldsymbol{h}, \boldsymbol{\lambda})$, *in the sense that* $\max_{\boldsymbol{\lambda} \in [0,B]^J} \hat{\mathcal{L}}(\bar{\boldsymbol{h}}, \boldsymbol{\lambda}) - \min_{\boldsymbol{h} \in \mathrm{conv}(\mathcal{H})} \hat{\mathcal{L}}(\boldsymbol{h}, \bar{\boldsymbol{\lambda}}) \leq \nu$, *and* $\boldsymbol{h}^* \in \mathrm{conv}(\mathcal{H})$ *satisfies* $\mathcal{V}(\boldsymbol{h}^*) \leq 0$, *then*

$$\mathcal{E}(\bar{\boldsymbol{h}}) \leq \mathcal{E}(\boldsymbol{h}^*) + \nu + \mathcal{O}(\rho \gamma(n, \mathcal{H}, \delta)), \quad \|\mathcal{V}(\bar{\boldsymbol{h}})\|_\infty \leq \frac{1+\nu}{B} + \mathcal{O}(\rho \gamma(n_{\min}, \mathcal{H}, \delta)) + \varepsilon.$$

Thus, as long as we can find an arbitrarily good saddle point, which weighted ERM grants if $\mathcal{H}^{werm}$ is expressive enough while having finite VC dimension, then we obtain consistency. A saddle point can be found by running a gradient ascent algorithm on $\boldsymbol{\lambda}$ confined to $[0, B]^J$, which repeatedly computes $h^t = \operatorname{argmin}_{h \in \mathcal{H}} \hat{\mathcal{L}}(h, \boldsymbol{\lambda}^t)$; the final $(\bar{\boldsymbol{h}}, \bar{\boldsymbol{\lambda}})$ are the averages of the primal and dual variables computed throughout the algorithm.

Although Theorem 5.1 captures the spirit of the argument for the plugin algorithm, it only applies naturally to the weighted ERM algorithm. This is because the plugin algorithm is solving a subtly different minimization problem: it returns $h^t$ as the *population minimum, if the estimated regression function $\hat{\eta}$ replaces the true regression function*.

| | MinOracle$_{h \in \mathcal{H}}(\mathcal{L}(h, \boldsymbol{\lambda}^t), z^n)$ | Update$_t(\boldsymbol{\lambda}, \mathbf{v})$ |
|---|---|---|
| FairReduction | $H \circ \text{argmin}_{f \in \mathcal{F}} \hat{L}(f)$ | $B \frac{\exp(\log \boldsymbol{\lambda} - \eta_t \mathbf{v})}{B - \sum_{j=1}^M \lambda_j + \sum_{j=1}^M \exp(\log \lambda_i - \eta_t v_i)}$ |
| FairCOCO | $\text{plugin}(\hat{\boldsymbol{\eta}}, (\hat{\pi}_g)_{g \in \mathcal{G}_{\text{fair}}}, \psi, \Phi, \boldsymbol{\lambda}^t)$ | $\text{proj}_{[0,B]^M}(\boldsymbol{\lambda} + \eta_t \mathbf{v})$ |

Table 1: The oracles shown are plugin (6) and ERM on the reweighted $\hat{L}$ (7). $H = [\text{argmax}_{k \in [K]}(\cdot)_k]$ converts a function $\mathcal{X} \to \mathbb{R}^K$ to a classifier. In FairCOCO, $\hat{\boldsymbol{\eta}}$ is estimated from samples $z^{1:n/2} = \{(x_1, y_1), \ldots, (x_{n/2}, y_{n/2})\}$ and all of the other probability estimates $(\hat{\pi}_g)_g$ and $\{\widehat{\mathbf{C}}^g[h^t]\}_g$ are estimated from $z^{n/2:} = z^n \setminus z^{1:n/2}$.

**Theorem 5.2.** *With probability at least $1 - \delta$, if projected gradient ascent is run as* Update$_t(\boldsymbol{\lambda}, \mathbf{v}) = \text{proj}_{[0,B]^J}(\boldsymbol{\lambda} + \eta \mathbf{v})$ *for $T$ iterations with step size $\eta = \frac{1}{B\sqrt{T}}$ and for $t = 1, \ldots, T$, $h^t = \text{plugin}(\hat{\boldsymbol{\eta}}, (\hat{\pi}_g)_{g \in \mathcal{G}_{\text{fair}}}, \psi, \Phi)$, letting $\rho = \max\{\|\psi\|_1, \|\phi_1\|_1, \ldots, \|\phi_M\|_1\}$, $\rho_g = \sum_{j=1}^J \|\mathbf{V}_j^g\|_\infty$, $\rho_\mathcal{X} = \|\mathbf{D}\|_\infty + \sum_{j=1}^J \|\mathbf{U}_j\|_\infty$, $\Delta\boldsymbol{\eta} = \mathbb{E}\|\eta(x) - \hat{\boldsymbol{\eta}}(x)\|_1$, $\check{n} = \min_{g \in \mathcal{G}_{\text{fair}}} n_g$, then*

$$\kappa := \mathcal{O}\left(J\rho\sqrt{\frac{K^2 \log(\check{n}) + \log(\frac{|\mathcal{G}_{\text{fair}}|K^2}{\delta})}{\check{n}}}\right) + \Delta\boldsymbol{\eta}\left(\rho_\mathcal{X} + \sum_{g \in \mathcal{G}_{\text{fair}}} \frac{\rho_g}{\pi_g}\right) + \sqrt{\frac{\log(\frac{|\mathcal{G}_{\text{fair}}|}{\delta})}{n}} \sum_{g \in \mathcal{G}_{\text{fair}}} \frac{\rho_g}{\pi_g^2}$$

$$\implies \mathcal{E}_\psi(\bar{\boldsymbol{h}}^T) \leq \mathcal{E}_\psi^* + \frac{JB}{\sqrt{T}} + \mathcal{O}(BJ\kappa), \qquad \|\mathcal{V}_\phi(\bar{\boldsymbol{h}}^T)\|_\infty \leq \frac{2J}{\sqrt{T}} + \mathcal{O}(J\kappa).$$

A key point in the presented analyses (for both procedures) is that the dominating statistical properties depend on the number of fairness groups. We note that $|\mathcal{G}_{\text{fair}}| \ll |\mathcal{G}_{\text{intersectional}}| = |\mathcal{A}|$ for the independent case, so this significantly improves results. More broadly, we conjecture that the statistical bounds depend on $\min(|\mathcal{G}_{\text{fair}}|, |\mathcal{G}_{\text{intersectional}}|)$, and leave the details to future work. We also note the statistical dependence on the size of the smallest group. This seems to be unavoidable, as we need an estimate of the group fairness violation in order to control it. To this end, group violations may be scaled by group size, which leads instead to a dependence on the VC dimension of $\mathcal{G}_{\text{fair}}$, improving statistical dependence with small groups at the cost of some fairness [15]. We expect that the bounds may be improved by a more refined analysis, or modified algorithms with stronger assumptions. We leave this detail to future work.

### 5.1 Additional Related Work

Recent work by Foulds et al. [10], Kearns et al. [15] and Hebert-Johnson et al. [13] were among the first to define and study intersectional fairness with respect to parity and calibration metrics respectively. Narasimhan [20] provide a plugin algorithm for group fairness and generalization guarantees for the unrestricted case. [19] considered Bayes optimality of fair binary classification where the sensitive attribute is unknown at test time, using an additional sensitive attribute regressor. Cotter et al. [7] provide proxy-Lagrangian algorithm with generalization guarantees, assuming proxy constraint functions which are strongly convex, and argue that better generalization is achieved by reserving part of the dataset for training primal parameters and part of the dataset for training dual parameters. Celis et al. [5] provide an algorithm with generalization guarantees for independent group fairness based on solving a grid of interval constrained programs; their and Narasimhan [20]'s work are most similar to ours.

## 6 Experiments

We consider demographic parity as the fairness violation, i.e., $\phi_{\text{DP}}^\pm = \pm(\mathbf{C}_{0,1}^g + \mathbf{C}_{1,1}^g - \mathbf{C}_{0,1} + \mathbf{C}_{1,1}) - \nu$, combined with 0-1 error $\psi(\mathbf{C}) = \mathbf{C}_{01} + \mathbf{C}_{10}$ as the error metric. All labels and protected attributes are binary or binarized. We use the following datasets (details in the appendix): (i) Communities and Crime, (ii) Adult census, (iii) German credit and (iv) Law school. In the appendix, we also provide training curves, and experiments with the equal opportunity [12] fairness constraint. Code is available[2].

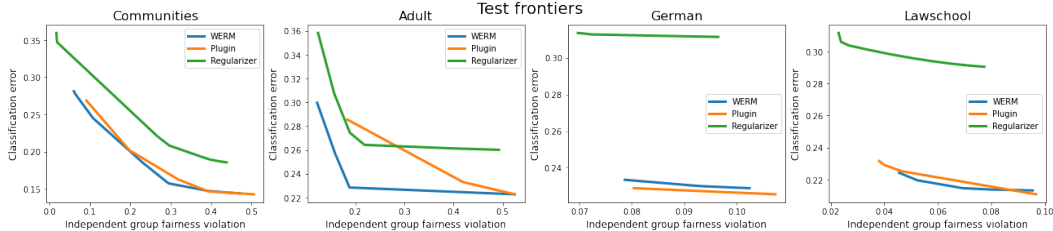

Figure 2: Experiments on independent group fairness, showing fairness frontier. The pareto frontier closest to the bottom left represent the best fairness/performance tradeoff.

Table 2: Average training times (averaged over the training sessions for each fairness parameter). The linear classifier with regularization is fastest. The Plugin Oracle is significantly faster than WERM.

|  | Independent | | | | Gerrymandering | | |
|  | C& C | Adult | German | Law school | Adult | German | Law school |
| --- | --- | --- | --- | --- | --- | --- | --- |
| Weighted-ERM | 107.0 s | 212.9 s | 16.5 s | 13.4 s | 817.0 s | 40.4 s | 49.4 s |
| Plugin | 11.0 s | 10.5 s | 6.3 s | 8.8 s | 699.8 s | 13.0 s | 17.7 s |
| Regularizer | 4.9 s | 4.5 s | 3.2 s | 4.6 s | N/A | N/A | N/A |
| Kearns et al. | N/A | N/A | N/A | N/A | 2213.7 | 821.5 s | 1674.4 s |

**Evaluation Metric.** We compute the "fairness frontier" of each method – that is, we vary the constraint level $\nu$. We plot the fairness violation and the error rate on the train set and a test set. The fairness violation for demographic parity is defined by

$$\text{fairviol}_{\text{DP}} = \max_{g \in \mathcal{G}_{\text{fair}}} |\widehat{\mathbf{C}}^g_{0,1} + \widehat{\mathbf{C}}^g_{1,1} - \widehat{\mathbf{C}}_{0,1} - \widehat{\mathbf{C}}_{1,1}|$$

Observe that on the training set, it is always possible to achieve extreme points by ignoring either the classification error or the fairness violation.

**Baseline:** `Regularizer` is a linear classifier implemented by using Adam to minimize logistic loss plus the following regularization function:

$$\rho \sum_{j=1}^{M} \left( \frac{\sum_{i:(z_i)_j=1} \sigma(w^\top x_i)}{|\{i : (z_i)_j = 1\}|} - \frac{\sum_{i=1}^{n} \sigma(w^\top x_i)}{n} \right)^2 \tag{3}$$

where $\sigma(r) = \frac{1}{1+e^{-r}}$ is the sigmoid function. This penalizes the squared differences between the average prediction probabilities for each group and the overall average prediction probability. Other existing methods we are aware of are either not applicable to overlapping groups, or are special cases of `GroupFair`.

**Experiment 1: Independent group fairness.** We consider independent group fairness, defined by considering protected attributes separately. Our results compare extensions of FairCOCO [20] and a FairReduction [1], existing special cases of `GroupFair` using the plugin and weighted ERM oracles respectively. Results are shown in Figure 2. We further present the differences in training time in 2. On all datasets, the variants of `GroupFair` are much more effective than a generic regularization approach. However, `Plugin` seems to violate fairness more often at test time – perhaps this is due to the $\|\hat{\eta} - \eta\|_1$ term in the generalization bound in Theorem 5.2. At the same time, `Plugin` is much faster (almost by an order of magnitude for two datasets), since its $\text{MinOracle}$ essentially has a closed-form solution, while `Weighted-ERM` has to solve a new ERM problem in each iteration.

**Experiment 2: Gerrymandering group fairness.** Unfortunately, intersectional fairness is not statistically estimable in most cases as most intersections are empty. As a remedy, [15] propose max-violation fairness constraints over $\mathcal{G}_{\text{gerrymandering}}$, where each group is weighed by group size, i.e., $\max_{g \in \mathcal{G}_{\text{gerrymandering}}} \frac{|g|}{n} |\widehat{\mathbf{C}}^g_{0,1} + \widehat{\mathbf{C}}^g_{1,1} - \widehat{\mathbf{C}}_{0,1} - \widehat{\mathbf{C}}_{1,1}|$, so empty groups are removed, and small groups have relatively low influence unless there is a very large fairness violation. We denote the approach of Kearns et al. [15] as `Kearns et al.` This approach is closely related to `Weighted-ERM` but searches for the maximally violated group by solving a cost-sensitive classification problem and uses fictitious play between $\lambda$ and $h$. For the `Plugin` and `Weighted-ERM` approaches, we optimize the cost function directly using gradient ascent, precomputing the gerrymandering groups present in the

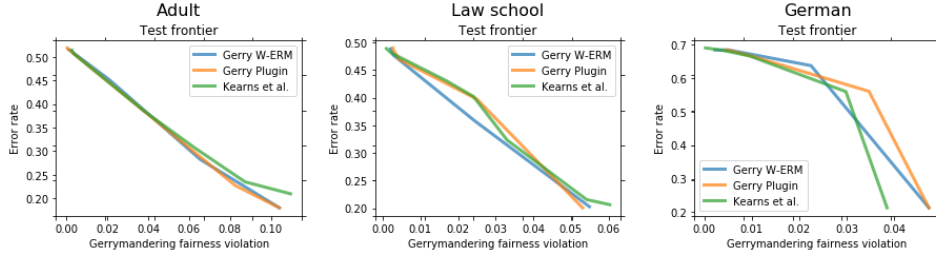

Figure 3: Experiments on gerrymandering group fairness. The pareto frontier closest to the bottom left represent the best fairness/performance tradeoff.

data. Results are shown in Figure 5. We further present the differences in training time in Table 2. The results are roughly equivalent in terms of performance, however, both the `Weighted-ERM` and `Plugin` approach are 1-2 orders of magnitude faster than `Kearns et al`.

## 7 Conclusion

This manuscript considered algorithmic fairness across multiple overlapping groups simultaneously. Using a probabilistic population analysis, we present the Bayes-optimal classifier, which motivates a general-purpose algorithm, `GroupFair`. Our approach unifies a variety of existing group-fair classification methods and enables extensions to a wide range of non-decomposable multiclass performance metrics and fairness measures. Future work will include extensions beyond linear metrics, to consider more general fractional and convex metrics. We also wish to explore more complex prediction settings beyond classification.

## Broader Impact

This work has the following potential positive impact in the society: we hope that our work enables algorithmic fairness for intersectional groups, which most existing works ignore. At the same time, this work may have some negative consequences because it does not solve the problem of limited statistical power for severely under-represented minorities. This work also assumes that algorithmic fairness is appropriate, which may not always be the case. Furthermore, we should be cautious of the result of failure of the system when the assumed parametric definition for the fairness violation is unknown or not appropriate, yet generic metrics are used in its place.

## Footnotes

[2] https://github.com/frstyang/fairness-with-overlapping-groups

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
