[Supplementary Material]

**Theorem 3.1.** Under Assumption 2.1 and Assumption 3.1, if (1), i.e.,

$$\boldsymbol{h}^* \in \operatorname{argmin}_{\boldsymbol{h}} \ \mathcal{E}(\boldsymbol{h}) \text{ s.t. } \mathcal{V}(\boldsymbol{h}) \leq \mathbf{0},$$

is feasible (i.e., a solution exists), the Bayes-optimal classifier is given by $\mathbf{h}^*(\mathbf{x}) = \mathbf{h}^*(\mathbf{z}, \mathbf{a}) = \beta_{\mathbf{a}} \mathbf{h}_1(\mathbf{x}) + (1 - \beta_{\mathbf{a}}) \mathbf{h}_2(\mathbf{x})$, where $\beta_{\mathbf{a}} \in (0, 1), \forall \mathbf{a} \in \mathcal{A}$ and $\mathbf{h}_i(\mathbf{x})$ are weighted classifiers with weights $\{\{\mathbf{W}_{i,\mathbf{a}}\}_{i \in \{1,2\}}\}_{\mathbf{a} \in \mathcal{A}}$.

*Proof.* The key idea of the proof is to exploit the problem representation in terms of confusion matrices. The proof has two main steps (i) population analysis for feasible confusion matrices, and (ii) plug-in of the classifiers that achieve the Bayes optimal confusion.

**Confusion space.** As the first step, let $\mathcal{C}^g = \{\mathbf{C}^g(\mathbf{h}) \mid \mathbf{h} \in \mathcal{H}\}$ be all group $g$ specific confusion matrices, and let $\mathcal{C}_{\mathcal{G}_{\text{fair}}} = \prod_{g \in \mathcal{G}_{\text{fair}}} \mathcal{C}^g$ be the product space of all confusion matrices corresponding to fair groups associated with a given instance of the problem. Similarly, let $\mathcal{C}_{\mathcal{A}} = \prod_{g \in \mathcal{G}_{\text{intersectional}}} \mathcal{C}^g$ be the product space of all confusion matrices corresponding to intersectional groups. A standard property of confusion matrices is that each $\mathcal{C}^g$ is a convex set [21, 20, 24]. Thus, each $\mathbf{C} \in \mathcal{C}^g$ can be described as a mixture of two boundary points, i.e.,

$$\forall \mathbf{C} \in \mathcal{C}^g \, \exists \mathbf{C}^1, \mathbf{C}^2 \in \partial\mathcal{C}^g, \, \beta \in [0, 1], \text{ s.t. } \mathbf{C} = \beta\mathbf{C}^1 + (1 - \beta)\mathbf{C}^2$$

Another useful fact is that all confusion matrices on the boundary can be achieved by a weighted classifier [21, 20, 24]. This fact follows from the convexity of the set $C^g$, and is simply a dual representation – via support functions, i.e.,

$$\forall \mathbf{C} \in \partial\mathcal{C}^g, \, \exists \mathbf{W} \text{ s.t. } \mathbf{C} = \operatorname{Conf}^g(\mathbf{h}^*), \text{ where } \mathbf{h}^* \in \operatorname*{argmax}_{\mathbf{h} \in \mathcal{H}} \langle \mathbf{W}, \operatorname{Conf}^g(\mathbf{h}) \rangle,$$

and where, for notation clarity, we have $\operatorname{Conf}(\mathbf{h})$ as the confusion matrix of classifier $\mathbf{h}$, and $\operatorname{Conf}^g(\mathbf{h})$ as the group-restricted confusion matrix. Further, the solution $\mathbf{h}^*$ can be represented as a weighted classifier (Definition 3.2) [20, 24].

**Population confusion problem.** Recall that the population confusion can be decoposed into their intersectional counterparts $\mathbf{C} = \sum_{a \in \mathcal{G}_{\text{intersectional}}} \mathbb{P}(a)\mathbf{C}^a$. Similarly, each overlapping group confusion can be decomposed using the intersection confusions as $\mathbf{C}^g \in \mathcal{C}_{\mathcal{G}_{\text{fair}}}, \mathbf{C}^g = \sum_{a \in \mathcal{G}_{\text{intersectional}}} \mathbb{P}(a|g)\mathbf{C}^a$.

As the overall metric is a function of confusion matrices only, we can re-state (1) as the equivalent confusion problem (with slight abuse of notation) for any $\mathcal{G}_{\text{fair}}$ as:

$$\mathbf{C}^*, \{\mathbf{C}^{g,*}\} = \operatorname{argmin} \ \psi(\mathbf{C}) \text{ s.t. } \Phi(\mathbf{C}, \{\mathbf{C}^g\}) \leq \mathbf{0},$$

$$\mathbf{C} = \sum_{a \in \mathcal{G}_{\text{intersectional}}} \mathbb{P}(a)\mathbf{C}^a$$

$$\mathbf{C}^g = \sum_{a \in \mathcal{G}_{\text{intersectional}}} \mathbb{P}(a|g)\mathbf{C}^a$$

$$\mathbf{C}^a = \operatorname{Conf}^a(\mathbf{h}).$$

After substituting the population $\mathbf{C}$ and the group confusions $\mathbf{C}^g$ with the presented linear functions of $\mathbf{C}^a$, this is equivalent to the problem

$$\{\mathbf{C}^{a,*}\} = \operatorname{argmin} \ \psi(\{\mathbf{C}^a\}) \text{ s.t. } \Phi(\{\mathbf{C}^a\}) \leq \mathbf{0}, \quad \mathbf{C}^a = \operatorname{Conf}^a(\mathbf{h}).$$

Here, we have used the linearity of the cost functions $\psi$ and $\Phi$, and the linearity of the confusion matrix decompositions into intersectional confusion matrices.

**Putting it together.** The final step is noting that a solution, if it exists, can be represented by feasible intersectional confusion matrices $\{\mathbf{C}^{a,*}\}$, and in turn, each intersectional confusion matrix can be recovered as a weighted average of two intersectional boundary confusion matrices. Thus the corresponding classifiers can be recovered by a mixture of two weighted classifiers. $\square$

## A Independent vs. intersectional group fairness

**Proposition 3.2.** For any $\mathcal{G}_{\text{fair}}$ that satisfies assumption 2.1, suppose $\phi : [0, 1]^{K \times K} \times [0, 1]^{K \times K} \to \mathbb{R}_+$ is quasiconcave in its second argument, $\phi(\mathbf{C}, \mathbf{C}^g) \leq 0 \, \forall g \in \mathcal{G}_{\text{intersectional}} \implies \phi(\mathbf{C}, \mathbf{C}^g) \leq 0 \, \forall g \in \mathcal{G}_{\text{fair}}$. The converse does not hold.

*Proof.* (For the forward direction)

Recall that $f$ is *quasiconcave* if $f(\sum_i \lambda_i z_i) \leq \max_i\{f(z_i)\}$. When $\phi$ is quasiconvex, for any $\mathcal{G}_{\text{fair}}$, we can compute $\phi(\mathbf{C}, \mathbf{C}^g) = \phi(\mathbf{C}, \sum_{a \in \mathcal{G}_{\text{intersectional}}} \lambda_a \mathbf{C}^a) \leq \max_{a \in \mathcal{G}_{\text{intersectional}}} \phi(\mathbf{C}, \mathbf{C}^a)$, where $\lambda_a$ are linear weights (corresponding to inclusion probabilities).

Since $\phi(\mathbf{C}, \mathbf{C}^a) \leq 0$ by the claim, it follows that $\phi(\mathbf{C}, \mathbf{C}^a) \leq 0 \,\forall a \in \mathcal{G}_{\text{intersectional}} \implies \phi(\mathbf{C}, \mathbf{C}^g) \leq 0 \,\forall g \in \mathcal{G}_{\text{fair}}$. $\qquad\square$

**Converse.** Though the above applies to any quasiconcave metric, in this manuscript we mainly consider linear metrics. As a corollary, intersectional group fairness with respect to common fairness metrics such as demographic parity or equal opportunity implies independent group fairness. A simple xor-like example from [15] shows that the converse is not true.

We provide another counterexample to the converse, showing a gap between independent and intersectional demographic parity (DP) group fairness, on an example with more realistic structure.

**Example A.1.** Let $A_1, A_2, A_3$ be binary attributes and $\{A_m\}$ denote the event $\{A_m = 1\}$. If $\mathbb{P}(Y) = \mathbb{P}(A_1) = \mathbb{P}(A_2) = \mathbb{P}(A_3) = 0.5$, $A_1, A_2, A_3$ are both independent and conditionally independent given $Y$, and $\mathbb{P}(A_m \mid Y) = 0.6$, then for every $P, N \subset \{1, 2, 3\}$ with $P \cap N = \emptyset$

$$\mathbb{P}(Y \mid \cap_{i \in P} A_i, \cap_{j \in N} \bar{A}_j) = 0.5(1.2)^{|P|}(0.8)^{|N|}.$$

**Proposition A.1.** *An optimal (DP) intersectionally fair $\hat{Y}$ has, over every possible subgroup $G = \cap_{i \in P} A_i \cap_{j \in N} A_j$, $\mathbb{P}(\hat{Y} \mid G) = 0.384 = 0.5(1.2)^2(0.8)$ and has an error of $0.148$.*

*On the other hand, an optimal (DP) independently fair classifier has $\mathbb{P}(\hat{Y} \mid A_1, A_2, A_3) = 0.464$, $\mathbb{P}(\hat{Y} \mid A_i, A_j, \bar{A}_k) = 0.576$, $\mathbb{P}(\hat{Y} \mid A_i, \bar{A}_j, \bar{A}_k) = 0.384$, $\mathbb{P}(\hat{Y} \mid \bar{A}_i, \bar{A}_j, \bar{A}_k) = 0.656$ and has an error of $0.1$.*

Interestingly, even though $\mathbb{P}(Y \mid A_1, A_2, A_3) = 0.864$ and $\mathbb{P}(Y \mid \bar{A}_1, \bar{A}_2, \bar{A}_3) = 0.256$ have the highest and lowest probabilities, the reverse is true of the predictor $\hat{Y}$ – it sacrifices accuracy on these groups to obtain higher accuracy on mixed positive/complement intersections.

Here we set up and discuss the example in 3.2 in more detail. First we begin with a rigorous and more general description of the structure of the example – here, one can think of a binary attribute as being synonymous with a partition with two sections. The first section corresponds to individuals with a value of 1 for that attribute and the other section to those with a value of 0.

**Assumption A.2** (Independence). Assume that the binary attributes $A_1, A_2, \ldots, A_M$ and label $Y$ satisfy:

1. $A_1, \ldots, A_M$ are independent.

2. $A_1, \ldots, A_M$ are independent conditioned on $Y$.

In the following, when $A_j$ is used to denote an event inside a probability, it refers to the event $\{A_j = 1\}$. $\bar{A}_j$ refers to the event $\{A_j = 0\}$. We also use the notation $A_j = A_j^1$ and $\bar{A}_j = A_j^0$.

**Proposition A.2.** *For every $j = 1, \ldots, M$, define $q_j = P(A_j \mid Y)$ and $a_j = P(A_j)$. Then, under Assumption A.2, for any index set $J = \{j_1, j_2, \ldots, j_J\}$ and $(b_j)_{j \in J} \in \{0, 1\}^J$,*

$$P(Y \mid A_j^{b_j}, j \in J) = \prod_{k=1}^{J} \left(\frac{q_{j_k}}{a_{j_k}}\right)^{b_k} \left(\frac{1 - q_{j_k}}{1 - a_{j_k}}\right)^{1 - b_k},$$

*Proof.*

$$P(Y \mid A_{j_1}^{b_1}, \ldots, A_{j_J}^{b_J}) = \frac{P(Y, A_{j_1}^{b_1}, \ldots, A_{j_J}^{b_J})}{P(A_{j_1}^{b_1}, \ldots, A_{j_J}^{b_J})}$$

$$= P(Y) \prod_{k=1}^{J} \frac{P(A_{j_k}^{b_k} \mid Y, A_{j_1}^{b_1}, \ldots, A_{j_{k-1}}^{b_{k-1}})}{P(A_{j_k}^{b_k} \mid A_{j_1}^{b_1}, \ldots, A_{j_{k-1}}^{b_{k-1}})}$$

$$= P(Y) \prod_{k=1}^{J} \frac{P(A_{j_k}^{b_k} \mid Y)}{P(A_{j_k}^{b_k})}$$

$$= P(Y) \prod_{k=1}^{J} \left( \frac{q_{j_k}}{a_{j_k}} \right)^{b_k} \left( \frac{1 - q_{j_k}}{1 - a_{j_k}} \right)^{1 - b_k}.$$

The third line follows by independence, Assumption A.2. □

The idea behind the above proposition is that with the independence assumption A.2, the structure of $P(Y \mid A_1^{b_1}, \ldots, A_M^{b_M})$ is such that we have $P(Y)$ scaled either by $q_j/a_j$ or $(1 - q_j)/(1 - a_j)$ depending on whether we are in $A_j$ or $\bar{A}_j$. This in a sense makes the effects of protected attributes "pile on." If we assume WLOG that $q_j/a_j \geq 1$, then $(1 - q_j)/(1 - a_j) \leq 1$.

**Example A.3.** Suppose that $M = 3$, $P(Y) = 0.5$, and for every $j = 1, 2, 3$, $a_j = P(A_j) = 0.5$ and $q_j = P(A_j \mid Y) = 0.6$. (This is possible because for every $J$, $0 \leq P(Y \mid A_j, j \in J) \leq 1$, aka is a well defined probability.) Applying Proposition A.2 noting $\frac{q_j}{a_j} = 1.2$, $\frac{1 - q_j}{1 - a_j} = 0.8$,

$$P(Y \mid A_1) = P(Y \mid A_2) = P(Y \mid A_3) = 0.5 \cdot 1.2 = 0.6,$$
$$P(Y \mid \bar{A}_1) = P(Y \mid \bar{A}_2) = P(Y \mid \bar{A}_3) = 0.5 \cdot 0.8 = 0.4,$$
$$P(Y \mid A_1, A_2) = P(Y \mid A_1, A_3) = P(Y \mid A_2, A_3) = 0.5 \cdot (1.2)^2 = 0.72$$
$$\forall 1 \leq j, k \leq 3, \quad P(Y \mid A_j, \bar{A}_k) = 0.5 \cdot 1.2 \cdot 0.8 = 0.48$$
$$\forall 1 \leq j, k \leq 3, \quad P(Y \mid \bar{A}_j, \bar{A}_k) = 0.5 \cdot 0.8 \cdot 0.8 = 0.32$$
$$P(Y \mid A_1, A_2, A_3) = 0.5 \cdot (1.2)^3 = 0.864$$
$$\forall 1 \leq i, j, k \leq 3, \quad P(Y \mid A_i, A_j, \bar{A}_k) = 0.5 \cdot (1.2)^2 \cdot 0.8 = 0.576$$
$$\forall 1 \leq i, j, k \leq 3, \quad P(Y \mid A_i, \bar{A}_j, \bar{A}_k) = 0.5 \cdot 1.2 \cdot (0.8)^2 = 0.384$$
$$P(Y \mid \bar{A}_1, \bar{A}_2, \bar{A}_3) = 0.5 \cdot (0.8)^3 = 0.256$$

.

**Fact A.4.** Assuming Assumption A.2 and the accuracy metric, the optimal intersectionally fair predictor $\hat{Y}$ assigns the probabilities

$$\forall b \in \{0, 1\}^M, \; P(\hat{Y} \mid A_1^{b_1}, \ldots, A_M^{b_M}) = \text{wmedian}_A \left\{ P(Y) \prod_{j=1}^{M} \left( \frac{q_j}{a_j} \right)^{b_j} \left( \frac{1 - q_j}{1 - a_j} \right)^{1 - b_j} \right\}$$

where the weighted median $\text{wmedian}_A$ of a set of $2^M$ numbers $\{r_{b^1} \leq \ldots \leq r_{b^{2^M}} : b^i \in \{0, 1\}^M\}$ is

$$r_{b^{i^*}}, \; i^* = \min\{i \in \mathbb{N} : \sum_{k \geq i} P(A_1^{b_1^k}, \ldots, A_M^{b_M^k}) \geq 0.5\}.$$

*(Proof sketch).* By thinking about it (or taking subgradient of $\mathbb{E}|Y - \hat{Y}|$), since we have the freedom to pick any constant to be the one to assign to every $P(\hat{Y} \mid A_1^{b_1}, \ldots, A_M^{b_M})$, we get the weighted median formula. □

**Fact A.5.** In example A.3, using Fact A.4 (an) optimal intersectionally fair predictor assigns $P(\hat{Y} \mid A_1^{b_1}, A_2^{b_2}, A_3^{b_3}) = 0.384$ and has an error of

$$\frac{1}{8} \left( |0.864 - 0.384| + 3 \cdot |0.576 - 0.384| + |0.256 - 0.384| \right) = 0.148.$$

On the other hand, an optimal independently group fair predictor assigns

$$P(Y \mid A_1, A_2, A_3) = 0.5 \cdot (1.2)^3 = 0.464$$
$$\forall 1 \leq i, j, k \leq 3, \quad P(Y \mid A_i, A_j, \bar{A}_k) = 0.5 \cdot (1.2)^2 \cdot 0.8 = 0.576$$
$$\forall 1 \leq i, j, k \leq 3, \quad P(Y \mid A_i, \bar{A}_j, \bar{A}_k) = 0.5 \cdot 1.2 \cdot (0.8)^2 = 0.384$$
$$P(Y \mid \bar{A}_1, \bar{A}_2, \bar{A}_3) = 0.5 \cdot (0.8)^3 = 0.656.$$

This predictor has an error of $\frac{1}{8}(|0.864 - 0.464| + |0.256 - 0.656|) = 0.1$. This is strictly less than the optimal intersectional error $0.148$, i.e. there is a gap.

*Proof.* By basically the same argument as for the intersectional case, it is optimal to have $P(\hat{Y} \mid A_1) = P(\hat{Y} \mid \bar{A}_1)$ be the median of $P(Y \mid A_1), P(Y \mid \bar{A}_1)$. Now we just need to verify that $\hat{Y}$ as defined above is independently group fair.

$$P(Y \mid A_i) = \frac{1}{4}\left(P(Y \mid A_i, A_j, A_k) + P(Y \mid A_i, \bar{A}_j, A_k) + P(Y \mid A_i, A_j, \bar{A}_k) + P(Y \mid A_i, \bar{A}_j, \bar{A}_k)\right)$$

$$= \frac{1}{4}(0.464 + 2(0.576) + 0.384) = 0.5$$

$$P(Y \mid \bar{A}_i) = \frac{1}{4}\left(P(Y \mid \bar{A}_i, A_j, A_k) + P(Y \mid \bar{A}_i, \bar{A}_j, A_k) + P(Y \mid \bar{A}_i, A_j, \bar{A}_k) + P(Y \mid A_i, \bar{A}_j, \bar{A}_k)\right)$$

$$= \frac{1}{4}(0.576 + 2(0.384) + 0.656) = 0.5.$$

Since $i \in \{1, 2, 3\}$ is arbitrary independent group fairness is satisfied. $\square$

# B Consistency and Generalization

**Theorem 5.2.** With probability at least $1 - \delta$, if projected gradient ascent is run ($\text{Update}_t(\boldsymbol{\lambda}, v) = \text{proj}_{[0,B]^J}(\boldsymbol{\lambda} + \eta v)$) for $T$ iterations with step size $\eta = \frac{1}{B\sqrt{T}}$ and for $t = 1, \ldots, T$, $h^t = \text{plugin}(\hat{\boldsymbol{\eta}}, (\hat{\pi}_g)_{g \in \mathcal{G}_{\text{fair}}}, \psi, \Phi)$, letting $\rho = \max\{\|\psi\|_1, \|\phi_1\|_1, \ldots, \|\phi_J\|_1\}$, then

$$\mathcal{U}_\psi(\bar{\boldsymbol{h}}^T) \leq \mathcal{U}_\psi^* + \frac{JB}{\sqrt{T}} + ((1+J)B + 1)\rho\left(4\sqrt{\frac{K^2 \log(2n_{\min})}{n_{\min}}} + \sqrt{\frac{\log(2(1 + |\mathcal{G}_{\text{fair}}|)K^2/\delta)}{n_{\min}}}\right)$$

$$+ \mathbb{E}\|\boldsymbol{\eta}(x) - \hat{\boldsymbol{\eta}}(x)\|_1 B\left(\rho_{\mathcal{X}} + \sum_{g \in \mathcal{G}_{\text{fair}}} + \frac{\rho_g}{\pi_g}\right) + 2\sqrt{\frac{\log(|\mathcal{G}_{\text{fair}}|/\delta)}{n}} \sum_{g \in \mathcal{G}_{\text{fair}}} \frac{\rho_g B}{\pi_g^2}$$

$$\|\mathcal{V}_\Phi(\bar{\boldsymbol{h}}^T)\|_\infty \leq \frac{2J}{\sqrt{T}} + 4(4(1+J) + 1)\rho\left(\sqrt{\frac{K^2 \log(2n_{\min})}{n_{\min}}} + \sqrt{\frac{\log(2(|1 + |\mathcal{G}_{\text{fair}}|)K^2/\delta)}{n_g}}\right)$$

$$+ 4\mathbb{E}\|\boldsymbol{\eta}(x) - \hat{\boldsymbol{\eta}}(x)\|_1\left(\rho_{\mathcal{X}} + \sum_{g \in \mathcal{G}_{\text{fair}}} \frac{\rho_g}{\pi_g}\right) + 8\sqrt{\frac{\log(|\mathcal{G}_{\text{fair}}|/\delta)}{n}} \sum_{g \in \mathcal{G}_{\text{fair}}} \frac{\rho_g}{\pi_g^2}.$$

*Proof.* First step is to extract the error incurred by plugging in $\hat{\boldsymbol{\eta}}$ rather than $\boldsymbol{\eta}$. Denoting $\hat{h} = \text{plugin}(\hat{\boldsymbol{\eta}}, (\hat{\pi}_g)_g, \psi, \Phi, \boldsymbol{\lambda})$ and $n_g = |\{i : x_i \in g\}|$ so that $\hat{\pi}_g = \frac{n_g}{n}$,

$$\hat{h}(x) = \text{argmin}_{k \in \{1, \ldots, K\}}\left\{\hat{\boldsymbol{\eta}}(x)^\top\left[\mathbf{D} + \sum_{l=1}^J \lambda_l\left(\mathbf{U}_l - \sum_{g \in \mathcal{G}_{\text{fair}}} \frac{\mathbb{1}_{x \in g}}{\hat{\pi}_g} \mathbf{V}_l^g\right)\right]\right\}_k.$$

Denote $h = \text{plugin}(\boldsymbol{\eta}, (\pi_g)_g, \psi, \Phi, \boldsymbol{\lambda})$. We quantify the discrepancy. Define $\hat{k} = \hat{h}(x)$ and $k^* = h(x)$. Also, define

$$\mathbf{M} = \mathbf{D} + \sum_{l=1}^J \lambda_l\left(\mathbf{U}_l - \sum_{g \in \mathcal{G}_{\text{fair}}} \frac{\mathbb{1}_{x \in g}}{\hat{\pi}_g} \mathbf{V}_l^g\right).$$

$$(\boldsymbol{\eta}(x)^\top \mathbf{M})_{\hat{k}} - (\boldsymbol{\eta}(x)^\top \mathbf{M})_{k^*} = (\hat{\boldsymbol{\eta}}(x)^\top \mathbf{M})_{\hat{k}} + [(\boldsymbol{\eta}(x) - \hat{\boldsymbol{\eta}}(x))^\top \mathbf{M}]_{\hat{k}} - (\boldsymbol{\eta}(x)^\top \mathbf{M})_{k^*}$$

$$\leq (\hat{\boldsymbol{\eta}}(x)^\top \mathbf{M})_{k^*} + [(\boldsymbol{\eta}(x) - \hat{\boldsymbol{\eta}}(x))^\top \mathbf{M}]_{\hat{k}} - (\boldsymbol{\eta}(x)^\top \mathbf{M})_{k^*} + \xi$$

$$= (\boldsymbol{\eta} - \hat{\boldsymbol{\eta}})^\top \mathbf{M}(e_{\hat{k}} - e_{k^*}) + \xi \leq \|\boldsymbol{\eta} - \hat{\boldsymbol{\eta}}\|_1 \left( \sum_{g \in \mathcal{G}_{\text{fair}}} \frac{\rho_g}{\pi_g} + \rho_{\mathcal{X}} \right) B + \xi$$

where $\rho_g = \sum_{l=1}^J \|\mathbf{V}_l^g\|_\infty$, $\rho_{\mathcal{X}} = \|\mathbf{D}\|_\infty + \sum_{l=1}^J \|\mathbf{V}_l\|_\infty$ and $\xi = 2\sqrt{\frac{\log(|\mathcal{G}_{\text{fair}}|/\delta)}{n}} \sum_{g \in \mathcal{G}_{\text{fair}}} \frac{\rho_g B}{\pi_g^2} -$
we are considering the fact that $|\pi_g - \hat{\pi}_g| \leq \sqrt{\frac{\log(2|\mathcal{G}_{\text{fair}}|/n)}{n}}$ for every $g \in \mathcal{G}_{\text{fair}}$ with probability $1 - \delta/2$. Taking expectation, we arrive at

$$\mathcal{L}(\mathbf{C}(\hat{h}), \boldsymbol{\lambda}) - \mathcal{L}(\mathbf{C}(h), \boldsymbol{\lambda}) \leq \mathbb{E}\|\boldsymbol{\eta}(x) - \hat{\boldsymbol{\eta}}(x)\|_1 \left( \sum_{g \in \mathcal{G}_{\text{fair}}} \frac{\rho_g}{\pi_g} + \rho_{\mathcal{X}} \right) B + 2\sqrt{\frac{\log(|\mathcal{G}_{\text{fair}}|/\delta)}{n}} \sum_{g \in \mathcal{G}_{\text{fair}}} \frac{\rho_g B}{\pi_g^2}. \tag{4}$$

By standard subgradient descent/online learning analysis, if the stepsize $\eta = 1/(B\sqrt{T})$ is used,

$$\frac{1}{T} \max_{\boldsymbol{\lambda} \in [0,B]^{2M}} \sum_{t=1}^T \hat{\mathcal{L}}(h^t, \boldsymbol{\lambda}) - \frac{1}{T} \sum_{t=1}^T \hat{\mathcal{L}}(h^t, \boldsymbol{\lambda}^t) \leq \frac{JB}{\sqrt{T}}$$

because $\mathcal{L}(h, \cdot)$ is concave and $\sqrt{J}$-Lipschitz (all fairness violations assumed to be in $[0, 1]$) and the $\ell_2$ radius of $[0, B]^J$ is $\sqrt{J}B$.

Now we show how good of a saddle point $\left( \frac{1}{T} \sum_{t=1}^T h^t, \frac{1}{T} \sum_{t=1}^T \boldsymbol{\lambda}^t \right) =: (\bar{\boldsymbol{h}}^T, \bar{\boldsymbol{\lambda}}^T)$ for the population problem. By convexity of $\mathcal{L}$ in the first argument,

$$\frac{1}{T} \max_{\boldsymbol{\lambda} \in [0,B]^M} \sum_{t=1}^T \hat{\mathcal{L}}(h^t, \boldsymbol{\lambda}) \geq \max_{\boldsymbol{\lambda} \in [0,B]^M} \hat{\mathcal{L}}(\bar{\boldsymbol{h}}^T, \boldsymbol{\lambda}).$$

Using equation 4 and the fact that $h^t$ is the minimizer of $\mathcal{L}(\mathbf{C}[h], \boldsymbol{\lambda}^t)$, but using $\hat{\boldsymbol{\eta}}$ instead of $\boldsymbol{\eta}$,

$$\frac{1}{T} \sum_{t=1}^T \hat{\mathcal{L}}(h^t, \boldsymbol{\lambda}^t) \leq \frac{1}{T} \sum_{t=1}^T \mathcal{L}(h^t, \boldsymbol{\lambda}^t) + \hat{\mathcal{L}}(h^t, \boldsymbol{\lambda}^t) - \mathcal{L}(h^t, \boldsymbol{\lambda}^t)$$

$$\leq \frac{1}{T} \sum_{t=1}^T \min_{h:\mathcal{X} \to [0,1]} \mathcal{L}(h, \boldsymbol{\lambda}^t) + \hat{\mathcal{L}}(h^t, \boldsymbol{\lambda}^t) - \mathcal{L}(h^t, \boldsymbol{\lambda}^t)$$

$$+ B(\rho_{\mathcal{X}} + \sum_{g \in \mathcal{G}_{\text{fair}}} \frac{\rho_g}{\pi_g})\mathbb{E}\|\boldsymbol{\eta}(x) - \hat{\boldsymbol{\eta}}(x)\|_1 + \xi$$

$$\leq \min_{h:\mathcal{X} \to [0,1]} \mathcal{L}(h, \bar{\boldsymbol{\lambda}}^T) + 4(1+J)B\rho \left( \sqrt{\frac{K^2 \log(K) \log(2n_{\min})}{n_{\min}}} + \sqrt{\frac{\log(2(1 + |\mathcal{G}_{\text{fair}}|)K^2/\delta)}{n_{\min}}} \right)$$

$$+ B(\rho_{\mathcal{X}} \sum_{g \in \mathcal{G}_{\text{fair}}} \frac{\rho_g}{\pi_g})\mathbb{E}\|\boldsymbol{\eta}(x) - \hat{\boldsymbol{\eta}}(x)\|_1 + \xi$$

where the middle term is from Lemma C.1. Let us absorb the error terms into $\gamma$. Now we can write:

$$\max_{\boldsymbol{\lambda} \in [0,B]^J} \hat{\mathcal{L}}(\bar{\boldsymbol{h}}^T, \boldsymbol{\lambda}) - \min_{h:\mathcal{X} \to [0,1]} \mathcal{L}(h, \bar{\boldsymbol{\lambda}}^T) \leq \frac{JB}{\sqrt{T}} + \gamma.$$

Letting $(\boldsymbol{h}^*, \boldsymbol{\lambda}^*)$ be primal dual optimal, we have

$$\forall \boldsymbol{\lambda} \in [0,B]^K, \quad \mathcal{L}(\boldsymbol{h}^*, \boldsymbol{\lambda}^*) \geq \hat{\mathcal{L}}(\bar{\boldsymbol{h}}^T, \boldsymbol{\lambda}) - \frac{JB}{\sqrt{T}} - \gamma. \tag{5}$$

The choices $\boldsymbol{\lambda} = 0$ and $\boldsymbol{\lambda} = \boldsymbol{\lambda}^* + \frac{B}{2}e_{g_m, \bullet}$ give

$$\hat{\mathcal{U}}(\bar{\boldsymbol{h}}^T) \leq \mathcal{U}(\boldsymbol{h}^*) + \gamma + \frac{JB}{\sqrt{T}}$$

$$\hat{\mathcal{V}}(\bar{\boldsymbol{h}}^T)_k \leq \frac{2}{B} \left( \frac{JB}{\sqrt{T}} + 2\gamma \right).$$

By Lemma C.1

$$\forall g \in \mathcal{G}_{\text{fair}}, \quad \sup_{h \in \mathcal{H}^{plg}} \|\mathbf{C}^g[h] - \hat{\mathbf{C}}^g[h]\|_\infty \leq 4\sqrt{\frac{K^2 \log(2n_g)}{n_g}} + \sqrt{\frac{\log(2(|1 + |\mathcal{G}_{\text{fair}}|)K^2/\delta)}{n_g}} =: \zeta(n_g).$$

we have that with probability $\geq 1 - \delta$

$$\mathcal{U}(\bar{\boldsymbol{h}}^T) \leq \mathcal{U}(\boldsymbol{h}^*) + \gamma + \frac{JB}{\sqrt{T}} + \rho\zeta(n_{\min})$$

$$\mathcal{V}(\bar{\boldsymbol{h}}^T)_k \leq \frac{2}{B}\left(\frac{JB}{\sqrt{T}} + 2\gamma\right) + \rho\zeta(n_{\min}).$$

Therefore we obtain the bounds

$$\mathcal{U}_\psi(\bar{\boldsymbol{h}}^T) \leq \mathcal{U}_\psi^* + \frac{JB}{\sqrt{T}} + ((1+J)B+1)\rho\left(4\sqrt{\frac{K^2\log(2n_{\min})}{n_{\min}}} + \sqrt{\frac{\log(2(1+|\mathcal{G}_{\text{fair}}|)K^2/\delta)}{n_{\min}}}\right)$$

$$+ \mathbb{E}\|\boldsymbol{\eta}(x) - \hat{\boldsymbol{\eta}}(x)\|_1 B\left(\rho_{\mathcal{X}} + \sum_{g \in \mathcal{G}_{\text{fair}}} + \frac{\rho_g}{\pi_g}\right) + 2\sqrt{\frac{\log(|\mathcal{G}_{\text{fair}}|/\delta)}{n}} \sum_{g \in \mathcal{G}_{\text{fair}}} \frac{\rho_g B}{\pi_g^2}$$

$$\|\mathcal{V}_\Phi(\bar{\boldsymbol{h}}^T)\|_\infty \leq \frac{2J}{\sqrt{T}} + 4(4(1+J)+1)\rho\left(\sqrt{\frac{K^2\log(2n_{\min})}{n_{\min}}} + \sqrt{\frac{\log(2(|1+|\mathcal{G}_{\text{fair}}|)K^2/\delta)}{n_g}}\right)$$

$$+ 4\mathbb{E}\|\boldsymbol{\eta}(x) - \hat{\boldsymbol{\eta}}(x)\|_1\left(\rho_{\mathcal{X}} + \sum_{g \in \mathcal{G}_{\text{fair}}} \frac{\rho_g}{\pi_g}\right) + 8\sqrt{\frac{\log(|\mathcal{G}_{\text{fair}}|/\delta)}{n}} \sum_{g \in \mathcal{G}_{\text{fair}}} \frac{\rho_g}{\pi_g^2}.$$

$\square$

## C Estimators

In this section, we give plugin and weighted ERM methods of solving the linear probabilistic minimization problems arising from the Lagrangian of our fairness problem. For clarity, we go over the choices of cost and constraint matrices corresponding to what we use in our experiments.

In our experiments, we maximize accuracy while enforcing independent demographic parity constraints and group-weighted gerrymandering demographic parity constraints. Under the framework of our probabilistic optimization problem, the former corresponds to the choice $\mathcal{G}_{\text{fair}} = \mathcal{G}_{\text{independent}}$, and $\Phi$ containing the $2|\mathcal{G}_{\text{independent}}| = 4M$ constraints

$$\forall g \in \mathcal{G}_{\text{independent}}, \pm(\mathbf{C}^g_{+,1} - \mathbf{C}_{+,1}) \leq \nu,$$

where the $+$ subscript denotes summing over indices $0, 1$ in place of $+$. I.e. for $g \in \mathcal{G}_{\text{indepdendent}}$, $\mathbf{V}^g_{g,\pm} = \pm\begin{bmatrix} 0 & 1 \\ 0 & 1 \end{bmatrix}$, $\mathbf{V}^{g'}_{g,\pm} = \mathbf{0}$ for $g \neq g'$, $\mathbf{U}_{g,\pm} = \pm\begin{bmatrix} 0 & 1 \\ 0 & 1 \end{bmatrix}$. $\mathbf{D} = \begin{bmatrix} 0 & 1 \\ 1 & 0 \end{bmatrix}$.

The latter corresponds to the choice $\mathcal{G}_{\text{fair}} = \mathcal{G}_{\text{gerrymandering}}$, and the $2|\mathcal{G}_{\text{gerrymandering}}|$ constraints

$$\forall g \in \mathcal{G}_{\text{gerrymandering}}, \pm\mathbb{P}(g)(\mathbf{C}^g_{+,1} - \mathbf{C}_{+,1}) \leq \nu.$$

This corresponds to, for $g \in \mathcal{G}_{\text{gerrymandering}}$, $\mathbf{V}^g_{g,\pm} = \pm\mathbb{P}(g)\begin{bmatrix} 0 & 1 \\ 0 & 1 \end{bmatrix}$, $\mathbf{V}^{g'}_{g,\pm} = \mathbf{0}$ for $g \neq g'$, $\mathbf{U}_{g,\pm} = \pm\mathbb{P}(g)\begin{bmatrix} 0 & 1 \\ 0 & 1 \end{bmatrix}$. The $\mathbb{P}(g)$'s will cancel out with the $\mathbb{P}(g)$'s in the expressions below.

## C.1 Plugin Estimator

Using linearity of $\psi$ and $\phi$, if $\eta$ is known, the population minimizer $h^* = \mathrm{argmin}_{h:\mathcal{X} \to [K]}\, \mathcal{L}(h, \lambda)$ is deterministic and has a convenient closed form solution (the same is true of any linear minimization).

$$
\mathcal{L}(h, \lambda) = \langle \mathbf{D} + \sum_{l=1}^{L} \lambda_l \mathbf{U}_l, \mathbf{C}[h] \rangle - \sum_{g \in \mathcal{G}_{\mathrm{fair}}} \sum_{l=1}^{L} \lambda_l \langle \mathbf{V}_l^g, \mathbf{C}^g[h] \rangle
$$

$$
= \mathbb{E}\bigg\{ \langle \mathbf{D} + \sum_{l=1}^{L} \lambda_l \mathbf{U}_l, \boldsymbol{\eta}(x) \boldsymbol{h}(x)^\top \rangle - \sum_{g \in \mathcal{G}_{\mathrm{fair}}} \sum_{l=1}^{L} \lambda_l \langle \mathbf{V}_l^g, \frac{\mathbb{1}_{\{x \in g\}}}{\mathbb{P}(g)} \boldsymbol{\eta}(x) \boldsymbol{h}(x)^\top \rangle \bigg\}
$$

$$
= \mathbb{E}\boldsymbol{\eta}(x)^\top \Big[ \mathbf{D} + \sum_{l=1}^{L} \lambda_l \Big( \mathbf{U}_l - \sum_{g \in \mathcal{G}_{\mathrm{fair}}} \frac{\mathbb{1}_{x \in g}}{\mathbb{P}(g)} \mathbf{V}_l^g \Big) \Big] \boldsymbol{h}(x).
$$

where we noticed that the conditional group confusion equals $\mathbf{C}^g[h] = \mathbb{E}\mathbb{1}_{\{x \in g\}} \boldsymbol{\eta}(x) \boldsymbol{h}(x)^\top / \mathbb{P}(g)$. Denote $\pi_g = \mathbb{P}(g)$ for $g \in \mathcal{G}_{\mathrm{fair}}$ as the group probabilities. Thus, the minimizer has the deterministic form

$$
h^*(x) = \mathrm{argmin}_{k \in \{1,\ldots,K\}} \bigg\{ \eta(x)^\top \Big[ \mathbf{D} + \sum_{l=1}^{L} \lambda_l \Big( \mathbf{U}_l - \sum_{g \in \mathcal{G}_{\mathrm{fair}}} \frac{\mathbb{1}_{x \in g}}{\mathbb{P}(g)} \mathbf{V}_l^g \Big) \Big] \bigg\}_k. \tag{6}
$$

Finally, since we do not actually have access to the true $\boldsymbol{\eta}$, we replace $\boldsymbol{\eta}$ with an estimated $\hat{\boldsymbol{\eta}}$.

## C.2 Weighted ERM

In the weighed ERM approach (referred to as cost-sensitive classification for the binary case [1]) we parametrize $h : \mathcal{X} \to [K]$ by a function class $\mathcal{F}$ of functions $: \mathcal{X} \to \mathbb{R}^{\mathbf{K}}$. The classification is the argmax of the predicted vector, $h(\mathbf{x}) = \mathrm{argmax}_j(\mathbf{f}(\mathbf{x})_j)$, so we denote the set of classifiers as $\mathcal{H}^{werm} = \mathrm{argmax} \circ \mathcal{F}$. For a standard classification problem with 0-1 error, minimizing the dataset error $\widehat{\mathrm{err}}[h] = \frac{1}{n} \sum_{i=1}^{n} \mathbb{1}_{\{h(\mathbf{x}_i) \neq y_i\}}$ is done by minimizing a surrogate loss $\ell : \mathbb{R}^K \times [K] \to \mathbb{R}_+$, e.g., using softmax cross-entropy, over the dataset, as $\hat{\mathbb{E}}\ell(\mathbf{f}(\mathbf{x}), y) = \frac{1}{n} \sum_{i=1}^{n} \ell(\mathbf{f}(\mathbf{x}_i), y_i)$. Then we take $h = \mathrm{argmax} \circ f$.

Let $\ell(\mathbf{s}) \in \mathbb{R}^k$ be the vector $\ell(\mathbf{s})_k = \ell(\mathbf{s}, k)$.

In an analogous manner, we would like to minimize the empirical metric defined by the Lagrangian using a surrogate loss, as

$$
\min_{h \in \mathcal{H}^{werm}} \hat{\mathcal{L}}(h, \boldsymbol{\lambda}) = \sum_{i=1}^{n} e_{y_i}^\top \Big[ \frac{1}{n} \mathbf{D} + \sum_{l=1}^{L} \frac{\lambda_l}{n} \Big( \mathbf{U}_l - \sum_{g \in \mathcal{G}_{\mathrm{fair}}} \frac{\mathbb{1}_{x_i \in g}}{n_g} \mathbf{V}_l^g \Big) \Big] \boldsymbol{h}(x_i).
$$

where $n_g = |\{i : x_i \in g\}|$, $g \in \mathcal{G}_{\mathrm{fair}}$ are the empirical sizes of each group. Notice it has the form

$$
\min_{h \in \mathcal{H}^{werm}} \sum_{i=1}^{n} \mathbf{w}_i^\top \boldsymbol{h}(x_i) = \sum_{i=1}^{n} s(\mathbf{w}_i) \frac{\mathbf{w}_i^\top}{s(\mathbf{w}_i)} \boldsymbol{h}(x_i), \qquad s(\mathbf{w}_i) = \frac{1}{K-1} \sum_{k=1}^{K} (\mathbf{w}_i)_k.
$$

If we interpret $\mathbf{1} - \frac{\mathbf{w}_i}{s(\mathbf{w}_i)}$ as a probability distribution over labels and $s(\mathbf{w}_i)$ as its weight, then we have $\min_h \tilde{\mathbb{E}}[(\mathbf{1} - \tilde{\boldsymbol{\eta}}(x))^\top \boldsymbol{h}(x)]$ where $\tilde{\mathbb{P}}(x_i) = \frac{s(\mathbf{w}_i)}{\sum_{i=1}^{n} s(\mathbf{w}_i)}$ and $\tilde{\eta}(x_i) = \mathbf{1} - \frac{\mathbf{w}_i}{s(\mathbf{w}_i)}$.

A priori, $\frac{\max_k (\mathbf{w}_i)_k}{s(\mathbf{w}_i)} \leq 1$, i.e. $\frac{\max_k (\mathbf{w}_i)_k}{\sum_{k=1}^{K} (\mathbf{w}_i)_k} \leq \frac{1}{n-1}$, may not hold. But, since shifting each entry of $w_i$ by the same amount does not change the initial optimization problem, we can add the constant amount $(n-1) \max_k (\mathbf{w}_i)_k - \sum_{k=1}^{K} (\mathbf{w}_i)_k$ to each entry of $w_i$, after which $\frac{\mathbf{w}_i}{s(\mathbf{w}_i)} \leq \mathbf{1}$.

If $\ell$ is a surrogate loss used to minimize the multiclass error, it is assumed that we can minimize $\mathbb{E}[(1 - \boldsymbol{\eta}(x))_{h(x)}]$ if we minimize $\mathbb{E}[\boldsymbol{\eta}(x)^\top \ell(f(x))]$ and take $h = \mathrm{argmax} \circ f$. Therefore, we can solve the weighted version by minimizing reweighted surrogate loss:

$$
\min_{f \in \mathcal{F}} \tilde{\mathbb{E}}[\tilde{\boldsymbol{\eta}}(x)^\top \ell(f(x))] \equiv \min_{f \in \mathcal{F}} \sum_{i=1}^{n} s(\mathbf{w}_i) \Big( \mathbf{1} - \frac{\mathbf{w}_i}{s(\mathbf{w}_i)} \Big)^\top \ell(f(x)) =: \hat{L}(f). \tag{7}
$$

This provides a convex surrogate for the original problem of minimizing the empirical Lagrangian.

**Lemma C.1** (Confusion matrix generalization). *Denote $n_g$ as the number of samples belonging to group $g$ for $g \in \mathcal{G}_{fair} \cup \{\mathcal{X}\}$. Then with probability at least $1 - \delta$,*

$$\forall g \in \mathcal{G}_{fair} \cup \{\mathcal{X}\}, \quad \sup_{h \in \mathrm{conv}\,\mathcal{H}} \|\mathbf{C}^g[h] - \widehat{\mathbf{C}}^g[h]\|_\infty \leq 4\sqrt{\frac{\mathrm{VC}(\mathcal{H})\log(n_g+1)}{n_g}} + \sqrt{\frac{\log((1+|\mathcal{G}_{fair}|)K^2/\delta)}{n_g}}.$$

*Proof.* By standard binary classification generalization [2], with probability at least $1 - \delta$,

$$\sup_{h \in \mathrm{conv}\,\mathcal{H}} \left| P(Y = i, h(X) = j \mid g) - \hat{P}(Y = i, h(X) = j \mid g) \right|$$

$$\leq 4\sqrt{\frac{\mathrm{VC}(\mathcal{H})\log(n_g+1)}{n_g}} + \sqrt{\frac{\log(1/\delta)}{n_g}}.$$

Then we take a union bound over $|\mathcal{G}_{\mathrm{fair}}|$ confusion matrices and $K^2$ entries per confusion matrix. $\qquad\square$

**Theorem C.2.** *Suppose $\psi : [0,1]^{K \times K} \to [0,1]$ and $\Phi : [0,1]^{K \times K} \times ([0,1]^{K \times K})^{\mathcal{G}_{fair}} \to [0,1]^L$ are $\rho$-Lipschitz w.r.t. $\|\cdot\|_\infty$. Recall $\hat{\mathcal{L}}(\boldsymbol{h}, \boldsymbol{\lambda}) = \hat{\mathcal{E}}(\boldsymbol{h}) + \boldsymbol{\lambda}^\top (\hat{\mathcal{V}}(\boldsymbol{h}) - \varepsilon\mathbf{1})$. Let $\gamma$ denote the bound in Lemma C.1 that applies to $\mathbf{C}$, $\gamma_g$ the bound that applies to $\mathbf{C}^g$, and denote $\gamma_{\mathcal{G}_{fair}} = \max_{g \in \mathcal{G}_{fair}} \gamma_g$. If $\varepsilon \geq \rho\gamma$ then with probability $1 - \delta$:*

*If $(\bar{\boldsymbol{h}}, \bar{\boldsymbol{\lambda}})$ is a $\nu$-saddle point of $\max_{\boldsymbol{\lambda} \in [0,B]^L} \min_{\boldsymbol{h} \in \mathrm{conv}\,\mathcal{H}} \hat{\mathcal{L}}(\boldsymbol{h}, \boldsymbol{\lambda})$, in the sense that $\max_{\boldsymbol{\lambda} \in [0,B]^L} \hat{\mathcal{L}}(\bar{\boldsymbol{h}}, \boldsymbol{\lambda}) - \min_{\boldsymbol{h} \in \mathrm{conv}\,\mathcal{H}} \hat{\mathcal{L}}(\boldsymbol{h}, \bar{\boldsymbol{\lambda}}) \leq \nu$, and $\boldsymbol{h}^* \in \mathrm{conv}\,\mathcal{H}$ satisfies $\mathcal{V}(\boldsymbol{h}^*) \leq 0$, then*

$$\mathcal{E}(\bar{\boldsymbol{h}}) \leq \mathcal{E}(\boldsymbol{h}^*) + \nu + 2\rho\gamma \tag{8}$$

$$\|\mathcal{V}(\bar{\boldsymbol{h}})\|_\infty \leq \frac{1+\nu}{B} + \rho\gamma_{\mathcal{G}_{fair}} + \varepsilon. \tag{9}$$

Thus, as long as we can find an arbitrarily good saddle point, which follows from weighted ERM if $\mathcal{H}^{werm}$ is expressive enough while having finite VC dimension, then we obtain consistency.

*Proof.* By Lemma C.1, with probability $1 - \delta$,

$$|\mathcal{E}(\boldsymbol{h}) - \hat{\mathcal{E}}(\boldsymbol{h})| \leq \rho\gamma, \qquad \|\mathcal{V}(\boldsymbol{h}) - \hat{\mathcal{V}}(\boldsymbol{h})\|_\infty \leq \rho\gamma_{\mathcal{G}_{\mathrm{fair}}}. \tag{10}$$

Therefore, $\hat{\mathcal{V}}(\boldsymbol{h}^*) \leq \varepsilon$. Using this feasibility to argue the first inequality below:

$$\hat{\mathcal{E}}(\bar{\boldsymbol{h}}) - \hat{\mathcal{E}}(\boldsymbol{h}^*) \leq \hat{\mathcal{E}}(\bar{\boldsymbol{h}}) - \hat{\mathcal{L}}(\boldsymbol{h}^*, \bar{\boldsymbol{\lambda}}) = \hat{\mathcal{L}}(\bar{\boldsymbol{h}}, 0) - \hat{\mathcal{L}}(\boldsymbol{h}^*, \bar{\boldsymbol{\lambda}}) \leq \nu.$$

Then (8) follows from (10) and triangle inequality. For the next part,

$$B(\hat{\mathcal{V}}(\bar{\boldsymbol{h}})_k - \varepsilon) = \hat{\mathcal{L}}(\bar{\boldsymbol{h}}, Be_k) - \hat{\mathcal{L}}(\boldsymbol{h}^*, \bar{\boldsymbol{\lambda}}) + \hat{\mathcal{E}}(\boldsymbol{h}^*) - \hat{\mathcal{E}}(\boldsymbol{h}) \leq \nu + 1.$$

This and (10) imply (9). $\qquad\square$

## D  Datasets

Here we dicsuss the datasets used and additional experimental details.

**Communities and Crime:** contains neighborhoods featurized by various statistics pertaining to the neighborhoods, e.g. percent employed in various professions, demographics, rent, etc. The label is whether there is a high ($> 70\%$-ile) rate of violent crimes per capita. There are $n = 1994$ samples and $N = 12$ protected attributes comprising various racial statistics.
**Adult census:** contains census data for $n = 2020$ individuals. The label is whether an individual has high income. $N = 7$ protected attributes comprising age, sex, and different races.
**German credit:** [8] contains features such as financial holdings, occupation, housing, and reason for purchases, and the goal is to predict whether an individual has good credit. Several categorical variables were converted to one-hot encodings. There are $n = 1000$ examples and $N = 3$ protected attributes corresponding to age, sex, and foreign worker status.
**Law school:** contains $n = 1823$ students and their gpas, cluster, and LSAT score. The goal is to predict whether the student passes the bar, and the protected attributes are age, gender, and family income.

For the constraint level $\nu$ we vary according a logarithmically spaced grid from $0.001$ to $1$ with $20$ points. We set $B = 50$ for the `GroupFair` methods. We vary the regularization parameter $\rho$ from $0.01/M$ to $1000/M$ across a logarithmically spaced grid with $20$ points.

The authors of [15] apply fictitious play to the gerrymandering problem, searching for the most violated constraint $\max_{g \in \mathcal{G}_{\text{fair}}} \frac{n_g}{n} |\mathbf{C}_{0,1}^g + \mathbf{C}_{1,1}^g - \mathbf{C}_{0,1} - \mathbf{C}_{1,1}|$ in response to the average of the predictors computed so far (if the violation exceeds $\nu$), and computing the minimizing predictor in response to the average of the dual variables obtained from the most violated constraints so far. On the other hand, we directly apply our `GroupFair` framework to their original cost function (see[15]) i.e., the problem of maximizing accuracy subject to $\forall g \in \mathcal{G}_{\text{fair}}, \frac{|g|}{n} |\mathbf{C}_{0,1}^g + \mathbf{C}_{1,1}^g - \mathbf{C}_{0,1} - \mathbf{C}_{1,1}| \leq \nu$. Both approaches aim to solve this problem.

Here are the full (training in addition to test) plots for the independent and gerrymandering experiments, as well as plots where we constrained the true positive rate to be equal across groups $\phi_{\text{EO}}^{\pm} = \pm \left( \frac{1}{\mathbb{P}(Y=1|g)} \mathbf{C}_{1,1}^g - \frac{1}{\mathbb{P}(Y=1)} \mathbf{C}_{1,1} \right) - \nu$, and measured this deviation in predicted probability for the equal opportunity fairness violation. Accordingly, for the Regularizer approach we changed the penalty to condition on the label being 1.

Figure 4: Experiments on independent group fairness. The pareto frontier closest to the bottom left represent the best fairness/performance tradeoff.

Figure 5: Experiments on gerrymandering group fairness. The pareto frontier closest to the bottom left represent the best fairness/performance tradeoff.

Figure 6: Experiments on equal opportunity. The `GroupFair` approaches appear to have more issues with generalization in this setting, which is essentially equivalent to a demographic parity constraint conditioned on the label being 1. Interestingly, the plugin approach does not generalize on the adult dataset, but the WERM approach does.