[Reviews · NeurIPS 2020]

Review 1

Summary and Contributions: The paper seeks to provide a unifying view of various existing approaches to group-fairness in classification. Edit post-rebuttal: I thank the authors for addressing my points in their response.

Strengths: Providing a unifying perspective of existing approaches is very valuable, especially around different interpretations of groups (intersecting / independent) as described in the paper.

Weaknesses: (1) The idea of the "group taxonomy" is nice, but it's not exhaustive - why these definitions? For example, we could define groups on arbitrary combinations of AND and OR operations of protected attributes (e.g. "women, over 50 or less than 20"). Additionally, the distinction between the second bullet ("intersectional") and fourth ("gerrymandering intersectional") seemed a bit weird - isn't it immediate that the latter is redundant? (I think this is the result in Prop 3.2., but I didn't understand it, see below). (2) It would be good to justify or explain the focus on average fairness violation as opposed to worst-case violations. I'm afraid this could be maliciously exploited. For example if I want to discriminate against a specific group S, I can "artificially" add many subsets of S complement to my collection of groups G, and it would seem that I'm doing very well (because the average violation is small due to the exponential number of subgroups I've added). (3) The paper itself is not clearly written, in my opinion. I found the notation very unfriendly. I think the submission would be greatly improved if the notation is made more clear and the paper is made more self contained (see more below).

Correctness: I couldn't verify most of the claims in this paper, as reflected in my confidence score.

Clarity: No, I found the paper to be very difficult to read. (1) It could be made much more "self contained" (in the appendix if this is a question of space) - for example there are many references to results from [21], such as in Def 3.2 and Thm 3.1., which I guess are obvious to the authors but I had a hard time following. At the very least, minimal intuition / explanation / precise reference in [21] is in order, in my opinion. (2) Some terms are not properly defined, e.g. G_fair appears in the statement of Prop. 3.2. but I have no idea what this refers to - it isn't defined in the "taxonomy" following Assumption 2.1.

Relation to Prior Work: Yes

Reproducibility: Yes

Additional Feedback: Minor points: - typos in line 45 and 148 - "for any multiclass classifier" in line 81: why is this here? eta(x) is independent of a particular classifier. - calibration is a common fairness notion that doesn't fall under the fairness metrics considered in this setup (so perhaps a qualifier to the statement in line 122 should be added) Higher-level points: - In the plugin approach, how do errors in the estimation of \eta(x) impact the results?


Review 2

Summary and Contributions: The literature on group fairness is mainly concerned with equalizing some statistical notion of fairness across different groups. An overlooked aspect is how to define the groups (based on one or more sensitive attribute, or whether groups are overlapping or not). The paper proposes a unifying framework for different choices of defining groups as well as proposing an algorithm with provable convergence guarantees. -------------------------------------------------------------------------------- I have read the other reviews and the rebuttal and there has been a good amount of discussion. I would have liked to see some more details about the importance of the framework and the value it adds to the current literature (other than being a general framework) in the rebuttal.

Strengths: This is a relevant paper to the NuerIPS community. The paper is generally well written and the claims are sound. It is nice to see a unifying algorithm for different choices of defining group fairness.

Weaknesses: While it is nice to see a generalization that allows many different ways of defining groups instead of committing to one, it is not clear to me that the contributions are significant. In particular, while the theoretical results are interesting, the techniques used are pretty standard. The results in section 3.1 are rather trivial. Moreover, I would like to see a more compelling experimental section (see below). I would also like to see a discussion in the rebuttal that clearly illustrates the additional gains from the unification framework compared to the previous work. This is marginal. I voted a 5 but this can be a 6 on a different day.

Correctness: The theoretical claims seem correct to me, though I did not check the appendix in great details. The experimental methodology is sound. I would have liked to see a more comprehensive set of experiments e.g. while the datasets used in the work are commonly used datasets in the algorithmic fairness literature they are too small to compare the performance of different algorithms in terms of their running time. I would also have liked to see a discussion of what would have happened if other notions of fairness such as equality of opportunity is used instead of statistical parity.

Clarity: The paper is really clearly written. Section 5 is the only exception where theorem statements can use a more concise writing.

Relation to Prior Work: The work unifies several previous works e.g. works on different ways of defining group fairness or unifying several statistical notions of fairness (which is known already) and the paper discusses these relationships very clearly.

Reproducibility: Yes

Additional Feedback: Minor comments: (1) The references should be sorted (see e.g. line 16) (2) There are some typos: e.g. Probabiistic in line 45, we refer **the** interested in line 145, and weoighted in line 148 (3) Why e_i is not bold given that it is a vector? (4) One complain about the prior work in the introduction is that "intersectional groups are often empty with finite samples." This work does not seem to address this issue either.


Review 3

Summary and Contributions: The paper proposes an approach to learn Bayes-optimal classifiers that satisfy group fairness. The proposed framework admits different group partitions, over which fairness constraints are defined, and any fairness metrics that can be written as a linear function of the confusion matrices. Using this framework, two ERM methods are studied, with theoretical analysis of their consistency.

Strengths: This work introduces a framework that unifies many existing methods on fair classification. Group fairness metrics such as demographic parity and equalized opportunity can be represented in this framework, and the definition of “groups” is general enough to include perhaps all commonly used ones. Moreover, this paper considers multiclass classification, which has been relatively unexplored in the fairness literature. This provides a language to crisply understand and compare existing methods, and also motivates new methods. The paper also provides rigorous theoretical analysis of the proposed algorithms, clearly stating its assumptions and consistency guarantees.

Weaknesses: Empirical evaluation does not really compare against prior work on fair classification. However, I understand that the focus of this work is not on the fairness/accuracy performance of a new method, but rather to introduce a framework that unifies much of existing work.

Correctness: The claims about how the proposed framework unifies existing fairness definitions and methods are correct, except a possible mistake (or my misunderstanding) about DP (see additional feedback below). The algorithm and consistency results appear to be sound, although proofs were not carefully checked.

Clarity: The paper is well written and organized, although discussion about related work is somewhat spread throughout the paper and confusing to keep track of. The assumptions and limitations of the proposed framework and theoretical results are described clearly and easy to find. Some small details were unclear: - Remark 3.3 about converse of Prop. 3.2 not applying to G_gerrymandering was not clear. Was it meant as: fairness on gerrymandering groups imply fairness on intersectional groups but not for reasons related to Prop. 3.2? - In Theorem 3.1, it was not clear what h_1 and h_2 represent and what their significance is.

Relation to Prior Work: Relation to prior works is discussed clearly, either about how they differ or how they fit into the proposed framework. A small suggestion I have is to have a separate related work section rather than discussions throughout the paper.

Reproducibility: Yes

Additional Feedback: - Line 102: “independent” and “intersectional” seem to be switched. - Demographic parity aims to equalize the rate of positive predictions between groups. I think this corresponds to C_{0,1} + C_{1,1} instead of C_{0,0} + C_{1,1}. - Theorem 5.1: n_g was not defined before. (I assume it is the size of group g?)


Review 4

Summary and Contributions: The paper has studied group-fair classification and presented a general procedure to learn classifiers under different group fairness constraints including independent and (Gerrymandering) intersectional groups. The paper proceeds with showing conditions under which plug-in and weighted ERM estimators are consistent. Finally, two experiments are presented --- in the independent group fairness case, weighted ERM and plug-in estimators outperform generic regularizer approach; in the Gerrymandering group fairness experiment, the performances of ERM and plug-in are similar to Kearns et al.

Strengths: The paper gives a nice unification of different group fairness treatment. It provides both a systematic way to solve the optimization problem as well as consistency guarantees of the empirical estimators given by the algorithm.

Weaknesses: - For theorem 5.1 and 5.2, is there a way to decouple the statement, i.e., separating out the optimization part and the generalization part? It would be clearer if one could give a uniform convergence guarantee first followed by how the optimization output can instantiate such uniform convergence. - In the experiments, is it reasonable for the German and Law school dataset to have shorter training time in Gerrymandering than Independent? Since in Experiment 2, ERM and plug-in have similar performance to Kearns et al. and the main advantage is its computation time, it would be good to have the code published.

Correctness: - Why do we need \epsilon (the generalization buffer) used in Algorithm 1? - For the experimental claims, see comments in the question above. - It's a bit hard to interpret the bounds, e.g., what the order of \rho and \rho_g is in theorem 5.2.

Clarity: Yes, the paper is well written in general. However, theorem 5.1 and 5.2 could be present in a clearer fashion, e.g., put the \nu-saddle point definition outside theorem 5.1 and describe the projected gradient ascent outside theorem 5.2. To compare theorem 5.1 and 5.2, is \nu of order O(1/\sqrt{T})? How should one interpret the many terms in \kappa? A bit more in-depth discussion after the propositions will also be helpful. Small Typo: - L148: weoighted -> weighted

Relation to Prior Work: In Section 2 (L103-118), the paper has discussed different group fairness notations. In Section 4.3 and Table 1, GroupFair can be seen as a generalization of the existing group fairness approaches like FairReduction and FairCOCO. In Section 5.1, more related works are listed. However, it will be more helpful to the reader if the distinction between this paper and Narasimhan [20] is made more explicit.

Reproducibility: No

Additional Feedback: There are many experimental details that should be included, e.g., the hyper-parameter used in Adam, how the train-test data is created, how the Kearns et. al. is run, etc. Since this is a theory paper, these details did not play an important factor when I evaluate the paper. However, it will be helpful for others who want to replicate the results in the paper if the authors can release their code upon publication of the paper.

[Author Response · NeurIPS 2020]

We thank the reviewers for providing helpful feedback and for pointing out typos, which we will fix in the final version.

**R2** and **R4** note that it would be desirable to have equality of opportunity experiments and code available. While our
paper is focused on algorithmically unifying and theoretically clarifying different notions of group fairness and their
statistical guarantees, rather than an extensive empirical comparison of different algorithms, we agree these would be
good to have and will provide them in the final version of the paper.

**Contributions of our framework.** First of all, we highlight independent group fairness, which arises naturally
from independently requiring fairness for each attribute. This is proposed in existing work e.g. [15], but to our
knowledge, we are first to analyze it. A key algorithmic contribution in this case is a plugin approach that keeps track
of a confusion matrix for each independent group instead of a confusion matrix for each intersectional group, as an
appropriate extension of [20] would have. This is enabled by the inverse group probability weighting (2) and shows
that the statistical dependence is *linear* instead of *exponential* in number of groups (which existing analyses suggest).
Juxtaposing different notions of fairness also leads one to study relationships between them, e.g., what is the nature
of intersectional fairness violations when independent fairness is enforced? We give an example in the appendix. For
probabilistic results, we shed light on Bayes optimal predictors in Section 3 and in our plugin algorithm, which are
new results. Finally, our framework has the benefit of generality, exactly capturing previous approaches as well as
insufficiently explored *multiclass* problems (previously unexplored, and nontrivial to extend to).

**Addressing specific comments by reviewers.**

- 18 **(R1)** Our framework is flexible and certainly allows one to use OR to define $\mathcal{G}_{\text{fair}}$, or any other grouping depen-
  19 dent on the sensitive attributes (Assumption 2.1); we simply chose the most common cases in practice/previous
  20 work (unrestricted, intersectional, independent, gerrymandering) for examples and experiments.

- 21 **(R1)** We emphasize that our paper does focus on *max-violation*, as the fairness problem we state places a
  22 constraint on each subgroup. Indeed, as the reviewer notes, the discrepancy between average and max-violation
  23 is the source of fairness gerrymandering, and is one of the motivations for this paper. The reason we use the
  24 term "average fairness violation" in Proposition 4.1 is that when the Lagrangian is formed for some choice
  25 of dual weights, the objective contains an average of the violations; but in the actual algorithm we minimize
  26 many different Lagrangians and combine them to produce a classifier which is fair for each group. We will
  27 clarify this possible misunderstanding in the final version.

- 28 **(R1)** Thank you for noting issues with understanding notation; which we believe are partially due to a lack of
  29 space in combination with the generality we set out to achieve. In the appendix, section C: Estimators walks
  30 through explicitly applying the plugin and W-ERM framework to an instance of the fairness problem. This
  31 may help, but does not fit in the main paper. We will include a more intuitive description of these methods in
  32 the main paper.

- 33 **(R1)** You are right in pointing out that calibration constraints are not linear functions of the confusion matrix;
  34 we'll make note of this in the final version.

- 35 **(R1)** The estimation error of eta appears in the middle term in the definition of $\kappa$ in Theorem 5.1.

- 36 **(R3)** Yes, the converse claim of Proposition 3.2 was that fairness *does not* imply intersectional fairness in
  37 general. Remark 3.3 was stating that gerrymandering fairness *does* imply intersectional fairness.

- 38 **(R3)** Theorem 3.1 states that an optimally fair classifier can be constructed from a convex combination of just
  39 two weighted classifiers, determined by some weight matrices $W_1, W_2$. This theorem serves to characterize
  40 the ideal solution to our problem.

- 41 **(R3)** You are right: DP should be $C_{0,1} + C_{1,1}$. The $C_{0,0} + C_{1,1}$ is a typo; in our experiments (and in the
  42 supplement, section C: Estimators) we used the correct definition. Oops.

- 43 **(R4)** Separating the generalization and optimization claims is ideal; this is essentially what we have done with
  44 Theorem 5.1, which isolates the generalization claim for the W-ERM approach. We had to give a separate
  45 statement, Theorem 5.2, for the plugin algorithm, because the algorithm involves updating $\boldsymbol{\lambda}$ with empirical
  46 violations but minimizing the Lagrangian with respect to the distribution defined by $\hat{\eta}$, which while learned
  47 from the empirical distribution is not quite the same. As a result, the proof has components that are hard to
  48 untangle.

- 49 **(R4)** Perhaps the discrepancy in times you have noticed is due to an old implementation of W-ERM based on
  50 the FairReduction code being used for the Independent experiments but not the Gerrymandering experiments.
  51 We will update this table in the final version.

- 52 **(R4)** The reason for the buffer $\varepsilon$ is that we would like to compare the classifier we find to the optimally fair
  53 classifier, but the optimally fair classifier may violate empirical finite sample fairness constraints.

- 54 **(R4)** The $\rho$ and $\rho_g$ constants are essentially $L_1$ norms, e.g. for DP all are between 2 and 4.

[Meta-Review · NeurIPS 2020]

The reviewers were overall in agreement that the paper provides a useful unified treatment of different notions of group fairness. The analysis of the multi-class classification setting also seems novel. However, the reviewers were unified in their concern that the independent group fairness definition is not well-motivated aside from computational considerations. Given that intersectional fairness has received considerable treatment in prior work (e.g., the fairness gerrymandering work), the authors should provide stronger practical motivation for the intermediate fairness definitions.